# LEARNING FROM PEERS IN REASONING MODELS

## ABSTRACT

Large Reasoning Models (LRMs) have the ability to self-correct even when they make mistakes in their reasoning paths. However, our study reveals that when the reasoning process starts with a short but poor beginning, it becomes difficult for the model to recover. We refer to this phenomenon as the *"Prefix Dominance Trap"*. This phenomenon indicates that the self-correction ability of LRMs is fragile and can be easily derailed by a poor start. This fragility motivates us to **look beyond internal self-correction**. Inspired by psychological findings that peer interaction can promote correction ability without negatively impacting already accurate individuals, we propose **Learning from Peers** (LeaP) to address this phenomenon. LeaP enables reasoning paths to periodically (every $T$ tokens) summarize and share intermediate reasoning via a routing mechanism, thereby incorporating peer insights. For smaller models that may inefficiently follow summarization and reflection instructions, we introduce fine-tuned **LeaP-T** models. Experiments on benchmarks including AIME 2024, AIME 2025, AIMO 2025, and GPQA Diamond demonstrate substantial improvements with LeaP. For example, QwQ-32B with LeaP achieves nearly 5 absolute points higher than its baseline on average and surpasses DeepSeek-R1-671B on three math benchmarks by an average of 3.3 points. The benefits of LeaP also generalize to other domains, such as logic puzzles on the ZebraLogic benchmark. Notably, our fine-tuned LeaP-T-7B matches the performance of DeepSeek-R1-Distill-Qwen-14B on AIME 2024. In-depth analysis reveals that LeaP provides robust error correction through timely peer insights and exhibits strong error tolerance. Code will be open-sourced.

## 1 INTRODUCTION

Large reasoning models (LRMs) (OpenAI, 2024a;b; 2025; Guo et al., 2025; Team, 2025) demonstrate strong performance on complex reasoning tasks. This success is largely attributed to their self-correction capability in test-time scaling (Zhang et al., 2025; Snell et al., 2024; Zeng et al., 2025b; Chen et al., 2024b), which consists of two emergent features: during generation, LRMs evaluate their current reasoning trajectories (self-verification) and may generate alternative ones (self-refinement) — a behavior often referred to as the "aha moment" (Guo et al., 2025). However, recent research reveals notable limitations in this self-correction mechanism. For example, researchers (Zeng et al., 2025b) observe that LRMs frequently become stuck in incorrect reasoning paths that are rarely corrected.

To assess LRMs' self-correction ability, we design a task where models must solve problems from fixed beginnings—drawn from both correct and incorrect responses. A powerful LRM should recover from poor starts and still reach the correct answer. Surprisingly, many LRMs, including QwQ-32B, suffer a nearly 20% performance drop when starting from a flawed beginning, even though it constitutes just 15% of the response length. We refer to this phenomenon—where *a short and flawed prelude can lead to a substantial degradation*—as the *"Prefix Dominance Trap"*. Unlike the well-recognized fact that models can make errors, our key insight is that their ability to self-correct is *fragile* and can be easily derailed by a poor start. This observation motivates us to look beyond internal self-correction, which most prior work (Huang et al., 2023) has focused on, and instead consider how external signals might provide the necessary nudge for recovery.

Psychological studies (Giuliodori et al., 2006; Snyder et al., 2015; Falk & Ichino, 2006) show that peer-based instruction helps students correct misconceptions effectively without harming those who are already correct. Inspired by this, we hypothesize that enabling LRMs to engage in peer learning may strengthen and extend their correction ability. Rather than being trapped by a poor prefix, models

can leverage insights from peers to escape error-prone trajectories. Building on this insight, we propose **Learning from Peers** (LeaP) to improve reasoning in large models: instead of relying solely on internal repair within a single reasoning path, **we enable LRMs to conduct communication and cross-path correction during parallel inference.** Concretely, during generation, after every $T$ tokens, each reasoning path summarizes its intermediate reasoning into a concise message, which is then shared with other paths through a heuristic routing mechanism. Simultaneously, each path receives summaries from its peers, creating opportunities for timely correction and robust reasoning.

We first validate our hypothesis by placing LeaP under the *"Prefix Dominance Trap"* setting. Experimental results show that models with LeaP reduce the performance gap by nearly 10% compared to those without LeaP. This finding suggests that LeaP encourages each reasoning path to verify not only its own trajectory but also those of its peers, thereby decreasing the risk of overlooking correct solutions. To further assess the effectiveness of LeaP, we conduct comprehensive evaluations **without any prefix** on AIME 2024 (MAA, 2024), AIME 2025 (MAA, 2025), the reference set of AIMO 2025 (Frieder et al., 2024), and GPQA Diamond (Rein et al., 2024). All reasoning models show significant improvement when using LeaP compared to those without LeaP, under comparable inference token budgets. Meanwhile, the results on ZebraLogic (Lin et al., 2025) demonstrate the generalization ability of LeaP.

However, during these experiments, we observe that smaller models without further training, such as DeepSeek-R1-Distill-Qwen-7B (Guo et al., 2025), sometimes struggle to follow summarization and reflection instructions effectively (Case studies in Appendix E.2). To address this, we propose **LeaP-T model which empowers LeaP with further training adaptation.** Our experiments show significant improvements. For example, the LeaP-T-7B model could achieve comparable performance with DeepSeek-R1-Distill-Qwen-14B (Guo et al., 2025) in AIME 2024.

Our contributions are: **I) Quantifying the *Prefix Dominance Trap***: We quantitatively validate the *Prefix Dominance Trap* in Large Reasoning Models (LRMs), demonstrating that even short, low-quality initial prefixes severely degrade performance. **II) A Parallel Inference Method with Cross-Path Correction:** We propose LeaP, which enables reasoning paths to communicate during inference. LeaP boosts QwQ-32B's performance by nearly 5 points on average, surpassing DeepSeek-R1-671B (Guo et al., 2025) on various benchmarks. We also comprehensively analyze its sensitivity to communication and robustness. **III) A Series of Trained Models:** We train and release a series of open-source models adapted to the LeaP framework to facilitate future research.

## 2 ENABLING CROSS-PATH CORRECTION IN PARALLEL INFERENCE: LEAP

### 2.1 MOTIVATION: PREFIX DOMINANCE TRAP

To assess the self-correction of LRMs, we introduce a task where LRMs are required to solve problems starting from fixed beginnings. If an LRM possesses strong correction abilities, it should more easily recover from reasoning paths that may lead to incorrect answers, thereby generating correct responses and resulting in a narrower accuracy gap with no-beginning. Specifically, we generate 32 responses per AIME 2024 (MAA, 2024) question using DeepSeek-R1-Distill-Qwen series (Guo et al., 2025) and QwQ-32B (Team, 2025). For each model, we select 10 incorrect responses from distinct questions, taking their

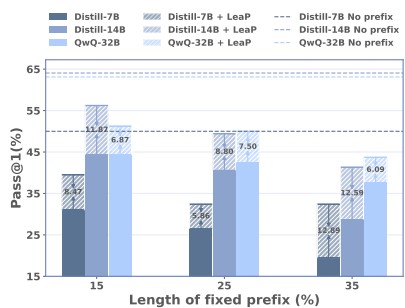

Figure 1: The results of starting with and without bad beginnings.

initial tokens from 15% to 35% as fixed prefixes. We then generate 16 continuations for each prefix using the respective LRM at temperature $\tau = 1$. Self-correction capability is assessed by the average accuracy gap ($P_G$) between these prefix-constrained and unconstrained generations. As reported with Pass@1 performance on AIME 2024 subsets (Figure 1), all tested LRMs exhibit a substantial performance drop (nearly 20% to 30%) when conditioned on these incorrect prefixes. This highlights **limited LRM self-correction ability**, a phenomenon we term the *"Prefix Dominance Trap"*.

### 2.2 METHODOLOGY: CROSS-PATH CORRECTION IN PARALLEL INFERENCE

To mitigate the *"Prefix Dominance Trap"*, we introduce Learning from Peers (LeaP), an inference-time strategy inspired by collaborative learning in psychology (Giuliodori et al., 2006; Snyder et al., 2015; Falk & Ichino, 2006). We hypothesize that structured peer interaction can extend an LRM's

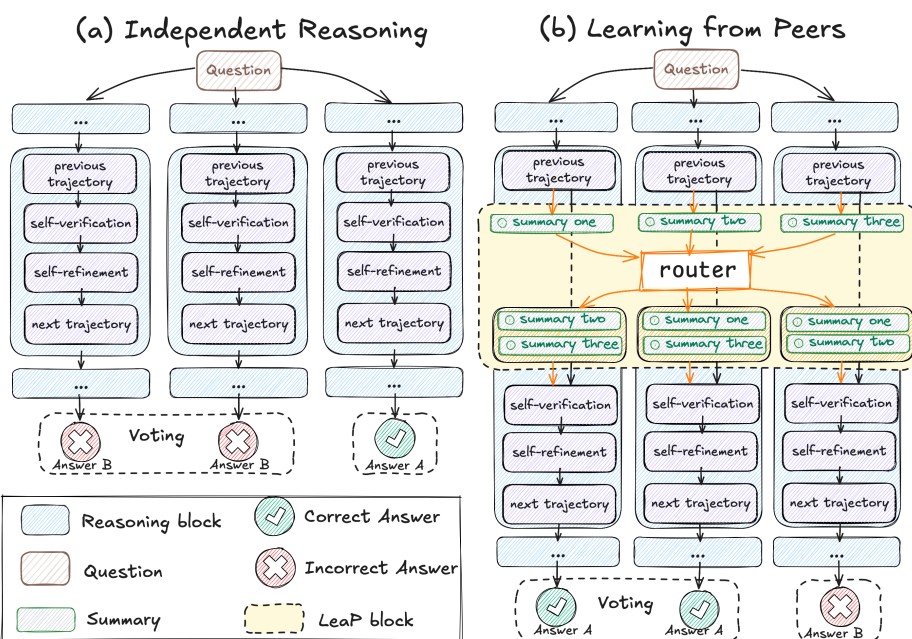

Figure 2: The illustration of (a) Independent Reasoning and (b) the proposed method Learning from Peers (LeaP). In independent reasoning, multiple paths are generated independently in parallel. In contrast, LeaP inserts a LeaP block into reasoning path, encouraging the model to learn from peers.

self-verification capabilities. Instead of relying on individual trial-and-error, reasoning paths leverage each other, broadening the search space for refinement. This shifts the LRM's focus from the demanding task of path generation to the simpler one of verifying and refining diverse existing paths, thereby reducing cognitive burden and enhancing reasoning effectiveness.

As depicted in Figure 2b, LeaP distinguishes itself from traditional independent reasoning (Figure 2a) by inserting *LeaP blocks* into the parallel inference process. Each block executes a two-stage procedure: (1) **Summarization**, where every path concisely reports its current state after every $T$ tokens, and (2) **Routing**, where each path incorporates the top-$k$ peer summaries selected by a routing strategy. Here, $T$ controls how frequently communication occurs, while $k$ controls how many peer insights are injected. An illustrative case of peer-triggered self-correction is shown in Figure 3.

**Stage I: Summarization.** Effective peer learning requires efficient insight sharing. The first stage in a LeaP block is Summarization. Each LRM path condenses its current approach, key insights, and intermediate results into a concise summary. This summary is strictly limited to 256 tokens to maintain token efficiency. A prompt, comprising a dynamically selected summary trigger and a summary template, directs this summarization. These elements are randomly chosen from predefined lists (see Appendix B), promoting variability in summary expression while ensuring core information capture. Once these concise summaries are generated by each path, the next step is to distribute them among peers. This is handled by the Routing stage.

**Stage II: Routing.** Although summaries provide condensed information, exposing each path to all peer summaries is overwhelming and token-inefficient, especially with numerous parallel paths. The Routing stage mitigates this by selecting which peer summaries a given path receives. For $N$ reasoning paths and their summaries $\{s_1, s_2, \ldots, s_N\}$, a routing function $\mathcal{R}$ selects for each path $i$ a top-$k$ subset of peer summaries $\mathcal{C}_i \subset \{s_j \mid j \neq i\}$. We explore three routing mechanisms, each employing a different heuristic to foster effective collaboration:

- **Dispersed routing:** Grounded in the **intuition** that diverse insights are crucial for breaking out of erroneous reasoning patterns and discovering novel solutions, Dispersed Routing prioritizes summaries that are least similar to the receiving path's own summary:

$$\mathcal{C}_i = \text{Top-}k\left((\texttt{dissimilarity}(s_i, s_j) \mid j \neq i)\right) \tag{1}$$

- **Clustered routing:** Conversely, the **assumption** here is that paths with similar reasoning are likely converging on a viable solution trajectory. Clustered Routing selects the top-$k$

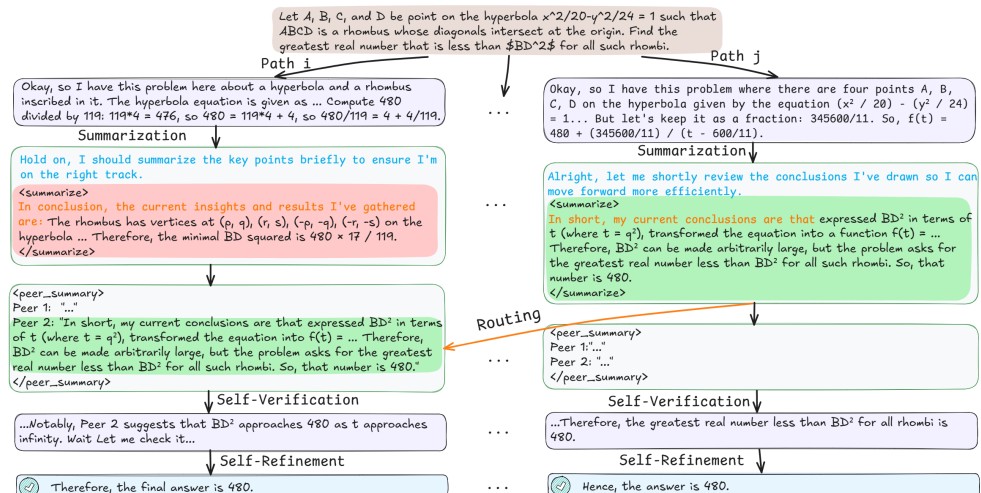

Figure 3: An example of how LeaP enables communication between path $i$ and $j$. Text in red indicates the current path is incorrect. Text in green shows a correct summary received from a peer.

most similar summaries, facilitating collaboration among paths that are already aligned:

$$\mathcal{C}_i = \text{Top-}k\left(\{\texttt{similarity}(s_i, s_j) \mid j \neq i\}\right) \quad (2)$$

- **Hybrid routing:** To achieve a synthesis of the above strategies, Hybrid Routing seeks to balance the collaborative reinforcement with the injection of diverse perspectives. It selects $\frac{k}{2}$ most similar summaries (exploitation) and $\frac{k}{2}$ most dissimilar summaries (exploration):

$$\mathcal{C}_i = \mathcal{C}_i^{\text{sim}} \cup \mathcal{C}_i^{\text{dis}}, \quad \text{where } |\mathcal{C}_i^{\text{sim}}| = |\mathcal{C}_i^{\text{dis}}| = \frac{k}{2} \quad (3)$$

To quantify the similarity between summaries $s_i$ and $s_j$ for these routing mechanisms, we employ the normalized Levenshtein similarity (Levenshtein et al., 1966). The similarity score $\texttt{similarity}(s_i, s_j)$ is defined as:

$$\texttt{similarity}(s_i, s_j) = 1 - \frac{D_{\text{lev}}(s_i, s_j)}{\max(|s_i|, |s_j|)} \quad (4)$$

where $D_{\text{lev}}(s_i, s_j)$ is the Levenshtein distance between $s_i$ and $s_j$, and $|s_i|$ denotes the length of the string $s_i$. This score, ranging from $[0, 1]$, provides a normalized measure where higher values indicate greater textual similarity. Notably, $\texttt{dissimilarity}(s_i, s_j) = 1 - \texttt{similarity}(s_i, s_j)$.

### 2.3 Can LeaP help in "Prefix Dominance Trap"?

We test LeaP's ability to overcome the *"Prefix Dominance Trap"* using the experimental setup from Section 2.1. As shown in Figure 1 (Pass@1 results on the AIME 2024 subset), LeaP consistently reduces the performance gap across all tested model sizes. Case studies in Appendix E.1 compare LeaP with the baseline. Furthermore, experiments with good initial prefixes, where beginnings likely lead to correct answers (Appendix C), also show LeaP outperforming independent reasoning. These results from both bad and good beginnings indicate that **LRMs leverage strong self-verification to assess peer reasoning paths**, thereby improving the correction ability of reasoning models.

## 3 Evaluating LeaP on Reasoning Benchmarks

To assess LeaP comprehensively, we evaluate the LeaP on four math/stem benchmarks and a language puzzle benchmark **without any fixed prefixes**. Section 3.1 describes the evaluation setup and implementation details. In Section 3.2, we report results showing the effectiveness of LeaP across four benchmarks. We assess the generalization ability of LeaP on a puzzle benchmark in Section 3.3.

### 3.1 Evaluation Setup

**Benchmarks and metrics.** We evaluate our method on challenging benchmarks. For mathematics, we use AIME 2024 (MAA, 2024) (30 problems), AIME 2025 (MAA, 2025) (30 problems), and the AIMO 2025 (Frieder et al., 2024) reference set (10 problems). Additionally, we test on the GPQA Diamond subset (Rein et al., 2024) (198 questions), its most difficult portion, demanding PhD-level expertise in physics, chemistry, and biology. We also use ZebraLogic (Lin et al., 2025), a challenging large-scale benchmark with over 3,000 complex natural language reasoning puzzles. For evaluation,

Table 1: We evaluate LeaP against the baseline on AIME 2024, AIME 2025, AIMO 2025, and GPQA Diamond, where our method shows significant outperformance. For comparison, we also report results for DeepSeek-R1-671B. As it lacks an official AIMO 2025 score, we evaluated it via four inference runs on the official website. R1-7B and R1-14B are the 7B and 14B versions of DeepSeek-R1-Distill-Qwen, respectively.

| Benchmarks | Models | Independent Reasoning | Self-Correct Prompt | Clustered | | Hybrid | | Dispersed | | R1-671B |
|---|---|---|---|---|---|---|---|---|---|---|
| | | | | Top-2 | Top-4 | Top-2 | Top-4 | Top-2 | Top-4 | |
| AIME 2024 | R1-7B | 51.35 | 52.81 | 56.15 | 59.27 | 59.17 | 61.67 | 60.31 | 60.52 | |
| | R1-14B | 64.47 | 65.63 | 72.08 | 71.04 | 71.77 | 74.48 | 71.15 | 77.29 | 79.8 |
| | QwQ-32B | 79.69 | 78.32 | 78.96 | 81.56 | 81.88 | 81.67 | 81.56 | 85.83 | |
| AIMO 2025 | R1-7B | 37.50 | 37.81 | 37.81 | 39.06 | 41.88 | 40.31 | 39.06 | 45.00 | |
| | R1-14B | 46.87 | 45.54 | 51.50 | 55.31 | 51.88 | 52.19 | 49.38 | 51.25 | 65.0 |
| | QwQ-32B | 63.75 | 64.38 | 63.75 | 65.63 | 64.06 | 64.69 | 67.19 | 67.19 | |
| AIME 2025 | R1-7B | 37.81 | 35.73 | 36.98 | 37.50 | 37.19 | 39.27 | 40.93 | 38.44 | |
| | R1-14B | 48.64 | 48.13 | 50.42 | 51.88 | 54.38 | 54.38 | 50.31 | 54.17 | 70.0 |
| | QwQ-32B | 68.13 | 68.96 | 68.85 | 71.35 | 71.04 | 72.50 | 70.83 | 71.67 | |
| GPQA Diamond | R1-7B | 46.91 | 47.66 | 52.97 | 52.65 | 51.83 | 53.47 | 53.28 | 55.56 | |
| | R1-14B | 53.47 | 53.09 | 54.80 | 58.33 | 54.42 | 57.89 | 55.68 | 55.05 | 71.5 |
| | QwQ-32B | 58.00 | 58.52 | 65.03 | 65.28 | 61.87 | 65.21 | 66.16 | 63.32 | |
| Avg. | R1-7B | 43.39 | 43.50 | 45.98 | 47.12 | 47.52 | 48.68 | 48.40 | 49.88 | |
| | R1-14B | 53.36 | 53.09 | 57.20 | 59.14 | 58.11 | 59.74 | 56.63 | 59.44 | 71.58 |
| | QwQ-32B | 67.39 | 67.55 | 69.15 | 70.96 | 69.71 | 71.02 | 71.44 | 72.00 | |

we use Pass@1 and Cons@N (Guo et al., 2025). We generate $N$ responses per question using a non-zero temperature. Pass@1 is Pass@1 $= \frac{1}{N}\sum_{i=1}^{N} \mathbb{I}_i$, where $\mathbb{I}_i$ indicates if the $i$-th response is correct. Cons@N denotes consensus results obtained by voting from $N$ responses.

**Models and baselines.** We use LRMs from 7B to 32B: the 7B and 14B versions of DeepSeek-R1-Distill-Qwen (Guo et al., 2025) and QwQ-32B (Team, 2025), all run on the vllm framework (Kwon et al., 2023). Our baselines are independent reasoning and self-correct prompt (Huang et al., 2023); for the latter, the model self-corrects after LeaP's summary stage instead of receiving peer insights. Following QwQ-32B (Team, 2025), we set generation parameters to a temperature of 0.6, Top-p of 0.95, and Top-k of 40. For Pass@1 scores, we set $N = 8$ for GPQA (Rein et al., 2024) and ZebraLogic (Lin et al., 2025), and $N = 32$ for math tasks (MAA, 2024; 2025; Frieder et al., 2024). Maximum tokens are 16,384 for 7B/14B models and 32,768 for the 32B model.

## 3.2 RESULTS

The Pass@1 results are presented in Table 1. Overall, LRMs with LeaP significantly outperform the baselines. For example, in terms of average performance across four benchmarks, our method using DeepSeek-R1-Distill-Qwen-7B with top-4 *Dispersed* routing exceeds the baseline by 6.49. Similarly, on DeepSeek-R1-Distill-Qwen-14B, it surpasses the baseline by 6.08. **Particularly, QwQ-32B with top-4 *Dispersed* routing even beats R1-671B in all three math datasets.**

We also observe performance improvements when transferring from Top-2 to Top-4 peer summaries. This trend suggests that as the number of top-$k$ peer summaries increases, the LRM benefits thinking paths from a greater number of peers. Among three routing strategies, *Dispersed*, and *Hybrid* clearly outperform the *Clustered* approach. This is expected, as receiving similar summaries from peers limits the overall diversity of trajectories for self-verification. In contrast, the diverse or complementary perspectives provided by *Dispersed* and *Hybrid* routing introduce a broader range of paths.

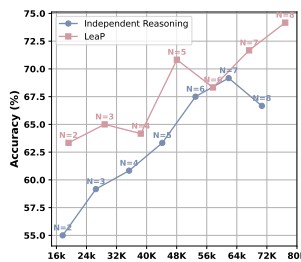

Figure 4: Cons@N vs. Total tokens on AIME 2024.

**Cons@N Results vs. Total tokens** To fairly compare LeaP with independent reasoning under matched compute budgets, we plot and analyze Cons@N as a function of the total tokens per question (including summaries and prompts) under the 7B size on AIME 2024. As the available compute (total tokens) grows, LeaP consistently outperforms independent reasoning (Cons@k) across the entire operating range (Figure 4). This demonstrates that LeaP is a more efficient use of compute: given the same token budget, cross-path correction yields higher consensus accuracy than simply increasing the number of independent samples. We also provide the detailed analysis of the computational overhead of LeaP in Appendix I.1.

### 3.3 CAN LEAP GENERALIZE TO OTHER NON-MATH/STEM TASKS?

To test LeaP's generalization beyond mathematical and scientific reasoning, we evaluated it on ZebraLogic (Lin et al., 2025), a benchmark of over 3,000 natural-language logical puzzles. Following the evaluation protocol in Section 3.1, we set LeaP with $T = 4096$, a Top-4 Dispersed Router, and report Pass@1 and Cons@8 to capture single-path quality and ensemble consensus.

Table 2: LeaP performance on ZebraLogic.

| Model | Pass@1 | Cons@8 |
|---|---|---|
| R1-7B | 46.88 | 53.11 |
| LeaP + R1-7B | **53.08** | **76.43** |
| R1-1.5B | 30.20 | 42.07 |
| LeaP + R1-1.5B | **37.45** | **62.04** |

These results in table 2 show that LeaP substantially improves both single-path accuracy (Pass@1) and ensemble consensus (Cons@8) on non-math logical reasoning tasks. This provides strong empirical evidence that the benefit of structured peer insights is not limited to numerical or STEM reasoning.

## 4 ADAPTING LEAP TO SMALL REASONING MODELS: LEAP-T

By analyzing the responses in Section 3.2, we observe that reasoning models with small sizes, such as DeepSeek-R1-Distill-Qwen-7B, sometimes fail to summarize and reflect on peers' summaries effectively. The case studies can refer to Appendix E.2. To this end, we introduce **LeaP-T** model series, where we attempt to alleviate this problem through supervised fine-tuning.

### 4.1 EXPERIMENTAL SETUP

We use approximately 1,000 AIME problems from 1984 to 2023 (MAA, 2024; 2025) as source data. We synthesize responses by applying LeaP to DeepSeek-R1-Distill-Qwen-32B (Guo et al., 2025) and filter suitable responses using a rule-based selection mechanism. We use supervised finetuning to train our LeaP-T models, starting from the 1.5B, 7B, and 14B versions of DeepSeek-R1-Distill-Qwen (Guo et al., 2025). Specifically, a response is selected if its final answer is correct and the length of all summaries is less than 256 tokens. We also generate SFT data without LeaP to train baseline models. Training hyperparameters are in Appendix F. We use the hyperparameters and metrics setting from Section 3.1. For LeaP, we use the Top-4 *Dispersed* routing and $T = 4096$ setting.

### 4.2 RESULTS

Table 3 presents Pass@1 and Cons@32 results for DeepSeek-R1-Distill-Qwen models (1.5B to 14B) fine-tuned with LeaP (termed LeaP-T) on three math benchmarks. LeaP-T consistently improves performance across model sizes; for instance, LeaP-T-1.5B achieves average gains of 4.45 in Pass@1 and 8.89 in Cons@32 over its baseline, with 7B and 14B models showing similar trends. LeaP-T models also outperform their counterparts where LeaP is applied without this specific fine-tuning. An ablation study reveals that further distilling these models (which are already distilled from R1-671B) using DeepSeek-R1-Distill-Qwen-32B via standard Supervised Fine-Tuning (SFT) without LeaP yields no improvement

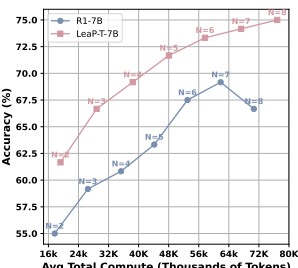

Figure 5: Cons@N vs. Total tokens on AIME 2024.

and sometimes slight degradation. This indicates our LeaP-T training effectively incorporates the *Learning from Peers* paradigm, rather than merely transferring knowledge through distillation.

We also assess the efficient test-time scaling ability of LeaP-T by comparing it with the baseline under the 7B size on AIME 2024 (MAA, 2024). As shown in Figure 5, we increase inference tokens by parallel generating multiple responses. For each point in Figure 5, we report average results from four repeated runs. It is clear that our LeaP-T-7B consistently outperforms DeepSeek-R1-Distill-Qwen-7B. This result demonstrates that our method scales more efficiently during test-time. We also compare our LeaP-T with MoA (Wang et al., 2024); details can be found in Appendix G.

## 5 IN-DEPTH ANALYSIS ON LEAP

To gain a deeper understanding of how and why LeaP works, we conduct a comprehensive analysis from three aspects: communication sensitivity (Section 5.1), robustness (Section 5.2), human verification (Section 5.3), computational overhead (Appendix I.1), homogenization (Appendix I.2) and the design of routers (Appendix I.3). This analysis not only guides the practical deployment of LeaP, but also sheds light on the inner workings of learning from peers.

Table 3: Evaluation of our LeaP-T from 1.5B to 14B on three math benchmarks.

| Models | AIME 2024 | | AIME 2025 | | AIMO 2025 | | Avg. | |
|---|---|---|---|---|---|---|---|---|
| | Pass@1 | Cons@32 | Pass@1 | Cons@32 | Pass@1 | Cons@32 | Pass@1 | Cons@32 |
| R1-1.5B | 32.00 | 50.00 | 24.69 | 30.00 | 14.00 | 30.00 | 23.56 | 36.67 |
| + SFT | 31.04 | **56.67** | 23.23 | 36.67 | 15.31 | 30.00 | 23.19 | 41.11 |
| + LeaP | 34.90 | 46.67 | 26.46 | 30.00 | 15.63 | 30.00 | 25.66 | 35.56 |
| LeaP-T-1.5B | **37.08** | **56.67** | **26.67** | **40.00** | **20.31** | **40.00** | **28.02** | **45.56** |
| R1-7B | 51.35 | 73.33 | 37.81 | 50.00 | 37.50 | 50.00 | 42.22 | 57.78 |
| + SFT | 51.56 | **80.00** | 35.73 | 53.33 | 33.75 | 40.00 | 40.35 | 57.78 |
| + LeaP | 60.52 | 76.67 | 38.44 | 53.33 | **45.00** | 50.00 | 47.99 | 60.00 |
| LeaP-T-7B | **64.38** | **80.00** | **41.25** | **56.67** | 44.06 | **60.00** | **49.90** | **65.56** |
| R1-14B | 64.47 | 80.00 | 48.64 | 60.00 | 46.87 | 60.00 | 53.33 | 66.67 |
| + SFT | 65.63 | **83.33** | 46.88 | 63.33 | 45.63 | 60.00 | 52.71 | 68.89 |
| + LeaP | **77.29** | **83.33** | 54.17 | 60.00 | 51.25 | 60.00 | 60.90 | 67.78 |
| LeaP-T-14B | 76.46 | **83.33** | **54.27** | **70.00** | **52.50** | 60.00 | **61.08** | **71.11** |

## 5.1 SENSITIVITY ANALYSIS OF COMMUNICATION

To better deploy LeaP in practice, we investigate the sensitivity of communication on several factors, including granularity $T$ (Section 5.1.1), traffic (Section 5.1.2), evolution tendency of types (Section 5.1.3), and position (Section 5.1.4).

### 5.1.1 ON COMMUNICATION GRANULARITY $T$

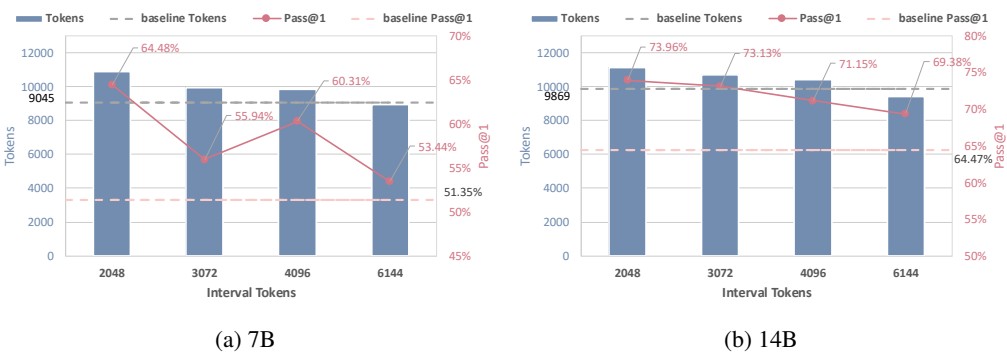

(a) 7B                                    (b) 14B

Figure 6: Pass@1 and total tokens on AIME 2024 for 7B and 14B models with LeaP, evaluated across interval tokens of LeaP from 2048 to 6144.

To study communication granularity $T$, we vary the $T$ between LeaP blocks for DeepSeek-R1-Distill-Qwen 7B and 14B models, using a fixed top-2 *Dispersed* routing strategy. Figure 6 shows that on AIME 2024, less frequent communication (larger $T$) slightly decreases Pass@1 performance. Specifically, as $T$ increases, Pass@1 for the 7B model drops from 64.48 to 53.44, and for the 14B model, from 73.96 to 69.38. Concurrently, larger $T$ (less frequent communication) consumes fewer tokens, as each LeaP block incurs token costs for summarization and routing. Overall, more frequent communication (smaller $T$) slightly improves performance but increases token usage, highlighting an accuracy-efficiency trade-off. Results on LeaP-T (Appendix I.4.2) further validate this finding.

### 5.1.2 ON THE COMMUNICATION TRAFFIC (TOP-K)

A common assumption is that increasing communication traffic (i.e., routing more summaries) improves performance by providing diverse information. However, our experiments indicate this intuition is not always correct. We investigate this by varying the number of routed summaries $k$ (from 1 to 16) for DeepSeek-R1-Distill-Qwen-14B on AIME 2024, using the *Dispersed* routing strategy and a fixed communication interval $T = 4K$ tokens. As shown in Figure 7, the Pass@1 score sharply increases from $k = 1$ to $k = 4$, peaking at $k = 4$. Subsequently, performance declines and fluctuates for $k > 4$. This suggests more communication traffic is not always beneficial. Insufficient information (small $k$) limits perspectives, while excessive summaries (large $k$) can introduce noise, overwhelm reasoning (particularly under token constraints), and impair coherence

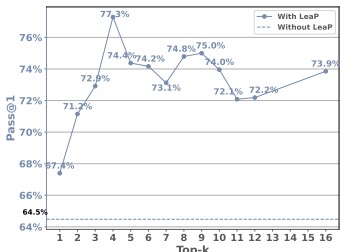

Figure 7: The top-$k$ Performance of LeaP-14B on AIME 2024.

and solution quality. Thus, a trade-off exists between communication richness and cognitive overload, with $k = 4$ (Top-4) providing the most effective balance.

### 5.1.3 ON EVOLUTION TENDENCY OF COMMUNICATION TYPES

Building on prior findings of optimal communication traffic, we analyze the evolution of communication *types* to understand peer learning dynamics. For this analysis, we fix the communication interval $T = 4K$ tokens and use the top-4 *Dispersed* routing strategy. We categorize communication outcomes as: *Consistent* (paths align with peers before and after communication), *Unaffected* (paths differ from peers and remain unchanged), or *Influenced* (paths initially differ but adjust after peer input). GPT-4o (OpenAI, 2024c) performs annotation, validated by humans on 10% of samples ($\approx$120) with over 95% agreement. Figure 8 plots this distribution for QwQ-32B on AIME 2024. Late in reasoning, *Unaffected* cases rise significantly. Conversely, the *Influenced* ratio peaks at 0.18 (at 8K tokens) and declines to 0.06 (at 24K tokens). This suggests communication is most impactful during early to mid-stage reasoning, with diminishing influence later.

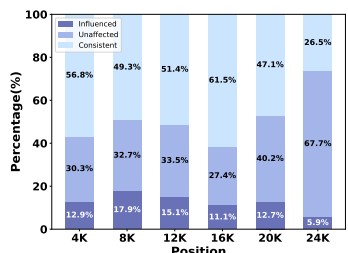

Figure 8: Communication types distribution of QwQ-32B on AIME 2024 at different positions, categorized into *Consistent*, *Unaffected*, and *Influenced*.

### 5.1.4 ON THE COMMUNICATION POSITION (WHEN ONLY COMMUNICATING ONCE)

**Setting on a new LeaP variant in single communication.** To further analyze sensitivity to communicating at different positions, we introduce a simplified variant where communication occurs only once during the reasoning process. This setting allows us to investigate the impact of communication position: whether it is more beneficial to communicate early, in the middle, or late. In Figure 9, we report results of DeepSeek-R1-Distill-Qwen-14B (Guo et al., 2025) on AIME 2024 (MAA, 2024) under this single communication setting. We observe that performance improves when communication occurs early in the reasoning process, increasing from 66.25 at 2K tokens to a peak of 69.48 at 4K tokens. However, beyond this point, performance declines, dropping to 65.33 by the end. These results suggest that early-stage communication is more effective than late-stage interaction. These findings are consistent with our previous conclusions from

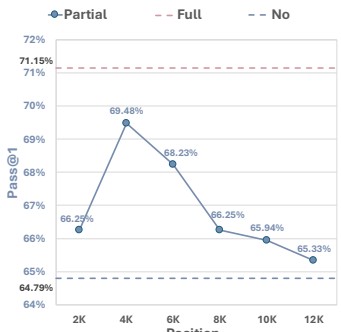

Figure 9: Performance of partial LeaP on the 14B model.

Section 5.1.3. Enabling a single LeaP block early in reasoning, a variant we term *LeaP-S*, yields substantial improvements over the baseline. Detailed results are in Appendix H.

### 5.2 POST-HOC ROBUSTNESS ANALYSIS OF LEAP

To better understand the reliability of LeaP in practical scenarios, we conduct a post-hoc robustness analysis. Specifically, we analyze whether the benefits of LeaP persist under two settings: (1) Vary the error path in peers (Section 5.2.1), and (2) Vary the difficulty levels (Section 5.2.2).

### 5.2.1 ROBUSTNESS ON ERROR TOLERANCE

A common concern is that low-quality paths in LeaP might mislead others, suggesting LeaP requires a high proportion of correct paths to avoid performance degradation from noise. To test LeaP's error tolerance, we vary the proportion of "good beginnings." For 10 questions, we select good and bad beginnings (initial 30% of

Figure 10: Pass@1 for various ratios of good beginnings, with beginning length at 30% of average response length.

tokens from DeepSeek-R1-Distill-Qwen-14B responses, following Section 2.1). We then construct 16 mixed initial responses per question, varying the good beginning proportion from 0% to 50%. Figure 10 reveals a surprising trend: LeaP consistently outperforms the baseline across all configurations, even with **no** good beginnings. For instance, with 0% good beginnings, LeaP's Pass@1 is 41.88, versus the baseline's 28.75. Remarkably, with only 43% good beginnings, LeaP surpasses the baseline's performance achieved when **all** its beginnings are good. These results counter the

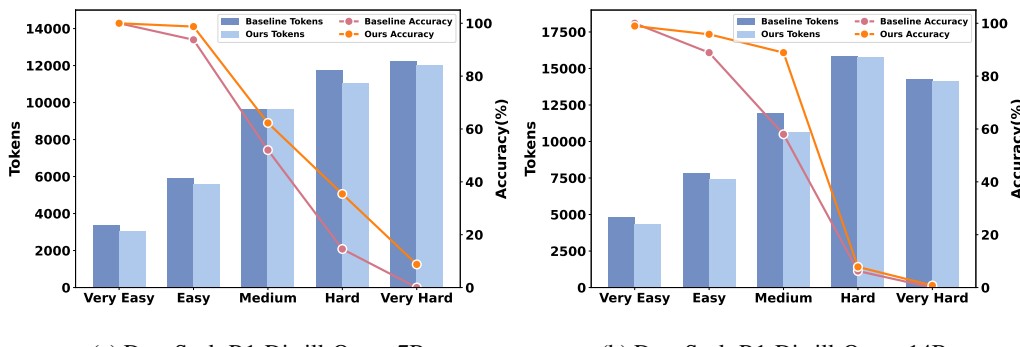

(a) DeepSeek-R1-Distill-Qwen-7B          (b) DeepSeek-R1-Distill-Qwen-14B

Figure 11: The Pass@1 and average token distribution across different difficulty levels, from Very Easy to Very Hard, for the 7B and 14B versions of DeepSeek-R1-Distill-Qwen.

assumption that LeaP requires mostly correct paths, demonstrating its high robustness. Peers can distill useful signals from noisy summaries, attributable to the self-verification ability of LRMs.

### 5.2.2 ROBUSTNESS AT VARIOUS DIFFICULTY LEVELS

Another concern is whether LeaP maintains robustness across various difficulty levels. It is possible that LeaP performs well primarily because the benchmark contains a certain proportion of simple questions. But can LeaP still provide benefits when problems become significantly more challenging?

To answer this question, we categorize AIME 2024 questions into five levels based on baseline correct responses (out of 32 from independent parallel reasoning): "Very Easy" (32 correct), "Easy" (25–31), "Medium" (9–24), "Hard" (1–8), and "Very Hard" (0). We then compute accuracy and reasoning-specific token usage (excluding peer summary tokens) per level. Figure 11 (results for DeepSeek-Distill-Qwen-7B and -14B) shows that **LeaP consistently improves accuracy across all difficulty levels**, even on "Very Hard" questions where the baseline fails completely. This suggests LeaP amplifies partial correctness and helps recover from complete failures. Furthermore, LeaP often uses fewer reasoning-specific tokens than the baseline. This, combined with LeaP's reduction of "Aha" moments (Section 3.2), indicates LeaP promotes earlier consensus and reduces overthinking. Appendix I.5 provides additional results for LeaP-T models, further confirming this robustness.

### 5.3 HUMAN EVALUATION ON THE 11ᵀᴴ PROBLEM OF AIME 2024

While benchmarks confirm LeaP's improved reasoning, they may not reveal underlying behavioral changes. For deeper insight, we performed a human evaluation comparing QwQ-32B's baseline and LeaP outputs on the 11th AIME 2024 problem. Figure 12 shows that on this problem, LeaP significantly increased correct responses from 25.00% (8/32) to 62.50% (20/32). We categorized response transitions to understand this improvement: *Correct → Correct*, *Incorrect → Correct*, *Incorrect → Incorrect*, and *Correct → Incorrect*. Notably, 40.62% of responses were *Incorrect → Correct*, showing LeaP repairs many flawed paths with peer communication. Importantly, no responses transitioned from *Correct → Incorrect*, suggesting communication rarely disrupts correct reasoning but primarily fixes errors. Case studies are in Appendix E.3.

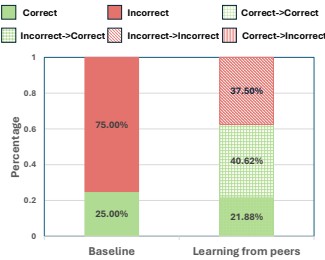

Figure 12: Human evaluation on the 11th problem of AIME 2024.

## 6 CONCLUSION

We identify the *"Prefix Dominance Trap"*, a phenomenon revealing the limited self-correction ability of large reasoning models (LRMs). To address this, we introduce Learning from Peers (LeaP), an approach where reasoning paths share insights, akin to note-passing in an exam. We also develop a series of trained models, LeaP-T, to empower this framework. Our experiments show LeaP significantly improves recovery from initial errors, often outperforming baselines and even larger models. In-depth analysis reveals the mechanisms behind these improvements, highlighting the potential of peer collaboration to enhance the reasoning abilities of large models.

ETHICS STATEMENT

We have reviewed the ICLR Code of Ethics and have carefully considered the potential ethical implications of our research. While Large Reasoning Models (LRMs) present broader ethical challenges, such as the potential for generating biased or factually incorrect content, our work is a focused empirical study on their reasoning capabilities using established public benchmarks. To the best of our knowledge, our proposed method does not introduce new ethical concerns or exacerbate existing ones. We believe our work is in full compliance with the ethical standards of the field.

REPRODUCIBILITY STATEMENT

To ensure the reproducibility of our research, we have included all necessary materials in our submission. The complete source code and the datasets used in our experiments are provided in the supplementary materials. Detailed information regarding hyperparameters, model configurations, and the required computational resources can be found in Appendix F.

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

# Appendix

## A    RELATED WORK

### A.1    SELF-CORRECTION BOTTLENECK

Recent studies show that while current Large Reasoning Models can generate lengthy reasoning chains, their performance does not consistently improve with increasing chain length Zeng et al. (2025b). In fact, longer reasoning chains often result in lower accuracy, which contradicts the assumption that extended reasoning improves problem-solving capabilities. Marjanović et al. (2025) further investigates the reasoning behavior of DeepSeek-R1, highlighting its tendency to dwell on previously explored problem formulations. This behavior hinders further exploration and leads to suboptimal reasoning paths. In this paper, we reaffirm their findings and present an effective method to address this issue.

### A.2    (INTERACTIVE) PARALLEL INFERENCE IN LLMs

Parallel inference Zeng et al. (2024); Zhang et al. (2025); Snell et al. (2024) enables LLMs to generate multiple reasoning paths simultaneously and aggregate them into a final answer. Self-Consistency (Majority Voting) Zeng et al. (2025b); Snell et al. (2024); Wang et al. (2022); Li et al. (2023); Brown et al. (2024); Song et al. (2024); Nguyen et al. (2024) selects the final answer by voting over candidate responses and choosing the one with the highest number of votes. Best-of-$N$ Gao et al. (2023); Cobbe et al. (2021); Sun et al. (2024); Gui et al. (2024); Amini et al. (2024); Sessa et al. (2024) improves response quality by generating multiple candidates and selecting the one with the highest reward. Another important direction involves interaction among multiple LLMs within collaborative frameworks during parallel inference. Multi-agent debate Du et al. (2023) establishes a symmetric discussion mechanism among agents. ReConcile Chen et al. (2023) and Corex Sun et al. (2023) treat collaboration as multi-round discussions or deliberations, using consensus mechanisms and role specialization (e.g., proposer, reviewer) to improve answer reliability. Building on these ideas, methods such as CoMM Chen et al. (2024a) and MALT Motwani et al. (2024) introduce explicit agent roles and diverse reasoning paths, enabling joint training for complex tasks. MoA Wang et al. (2024); Li et al. (2024) further proposes architectural hierarchies and network-based communication patterns to enhance collective reasoning. The key difference between our approach and methods like MoA lies in the **interaction mechanism between reasoning paths**. In MoA, each round only accesses the output from the previous round, without reference to earlier context. In contrast, our method enables **direct collaboration across multiple reasoning paths**, maintaining a finer-grained and more complete history of the reasoning process.

## B   PROMPTS

---

**Summarization Prompt Structure**

**Summary Trigger:** (randomly select one)

- Alright, let's take a step back and summarize what we've figured out so far.
- Wait, let me quickly recap what I've concluded so far.
- Alright, let me shortly review the conclusions I've drawn so I can move forward more efficiently.
- Hmm, a quick summary of what I've figured out might help streamline the next part of my reasoning.
- Hold on, I should summarize the key points briefly to ensure I'm on the right track.
- Okay, before continuing, let me put together a brief summary of the insights I've gathered so far.
- Okay, time to consolidate everything I've found into a concise summary.

**Summary Template:** (randomly select one)

- In short, my current conclusions are that ...
- To summarize, based on my previous reasoning, I have currently found that ...
- In conclusion, the current key takeaways and results are ...
- In short, I've currently concluded that ...
- To summarize, my recent findings are ...
- In conclusion, the current insights and results I've gathered are ...

---

Figure 13: The structure of the summarization prompt used during LeaP.

---

**Prompt for Different Tasks**

GPQA:
Please show your choice in the answer field with only the choice letter, e.g.,"ANSWER": "C".
Math Tasks:
Please reason step by step, and put your final answer within \boxed.

---

Figure 14: Prompts for different tasks.

---

**Prompt for Mixture-of-Agents**

Problem: {problem}
You have been provided with a set of responses from various open-source models to the latest user query. Your task is to synthesize these responses into a single, high-quality response. It is crucial to critically evaluate the information provided in these responses, recognizing that some of it may be biased or incorrect. Your response should not simply replicate the given answers but should offer a refined, accurate, and comprehensive reply to the instruction. Ensure your response is well-structured, coherent, and adheres to the highest standards of accuracy and reliability.
Responses from models:

---

Figure 15: Prompts for mixture of agents.

## C   PREFIX DOMINANCE TRAP FOR STARTING WITH GOOD BEGINNINGS

While the "Prefix Dominance Trap" highlights how poor beginnings can severely constrain reasoning, a natural follow-up question is whether good beginnings are sufficient to ensure correct final answers. To this end, we conduct a follow-up experiment similar to those in Section 2.1 under the *good*

*beginnings* setting, where the model start reasoning with a beginning of 15% average response length from correct reasoning paths.

As shown in Figure 16a, models initialized with good beginnings indeed perform better than random or bad initializations. This suggests that early correct cues provide useful guidance. However, these beginnings do not fully eliminate reasoning errors—many responses still deviate in later steps and arrive at incorrect conclusions. This observation reinforces that good beginnings can reduce—but not entirely prevent—reasoning failures.

We then examine how LeaP perform in the same setting. As illustrated in Figure 16b, our method consistently surpasses independent reasoning, even when starting from already high-quality beginnings. For instance, DeepSeek-Distill-Qwen-14B improves from 74.38 to 87.50 in Pass@1. This significant gain indicates that LeaP not only repairs faulty reasoning from poor prefixes but also mitigates subtle errors that emerge even when the reasoning starts correctly.

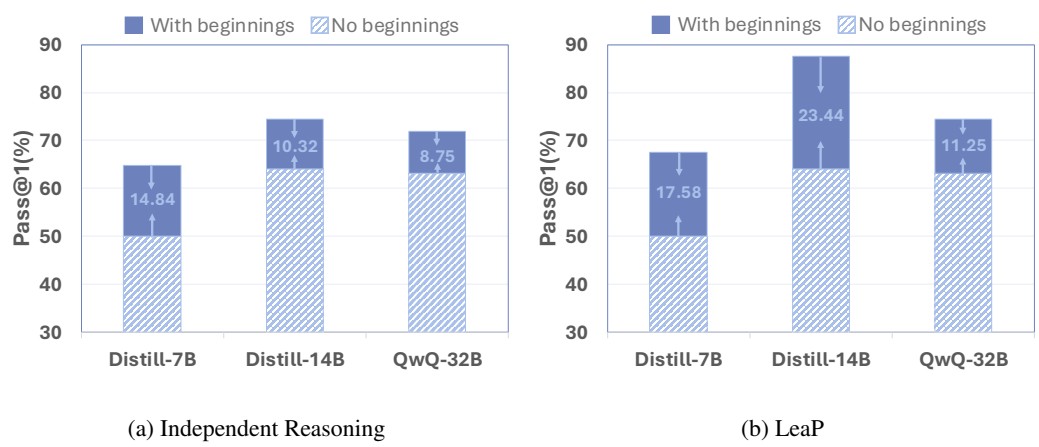

(a) Independent Reasoning                    (b) LeaP

Figure 16: The results of starting with a good beginning of 15% average response length.

## D    RANDOM ROUTING RESULTS

Table 4: The results of Pass@1 for four routing mechanisms on four benchmarks.

| Benchmarks | Models | Random | | Clustered | | Hybrid | | Dispersed | |
|---|---|---|---|---|---|---|---|---|---|
| | | Top-2 | Top-4 | Top-2 | Top-4 | Top-2 | Top-4 | Top-2 | Top-4 |
| AIME 2024 | R1-7B | 58.65 | 60.83 | 56.15 | 59.27 | 59.17 | 61.67 | 60.31 | 60.52 |
| | R1-14B | 72.60 | 75.83 | 72.08 | 71.04 | 71.77 | 74.48 | 71.15 | 77.29 |
| | QwQ-32B | 80.21 | 81.67 | 78.96 | 81.56 | 81.88 | 81.67 | 81.56 | 85.83 |
| AIMO 2025 | R1-7B | 40.62 | 42.81 | 37.81 | 39.06 | 41.88 | 40.31 | 39.06 | 45.00 |
| | R1-14B | 52.50 | 53.13 | 51.50 | 55.31 | 51.88 | 52.19 | 49.38 | 51.25 |
| | QwQ-32B | 63.75 | 68.44 | 63.75 | 65.63 | 64.06 | 64.69 | 67.19 | 67.19 |
| AIME 2025 | R1-7B | 38.02 | 39.17 | 36.98 | 37.50 | 37.19 | 39.27 | 40.93 | 38.44 |
| | R1-14B | 48.96 | 52.29 | 50.42 | 51.88 | 54.38 | 54.38 | 50.31 | 54.17 |
| | QwQ-32B | 71.35 | 71.98 | 68.85 | 71.35 | 71.04 | 72.50 | 70.83 | 71.67 |
| GPQA Diamond | R1-7B | 52.96 | 52.65 | 52.97 | 52.65 | 51.83 | 53.47 | 53.28 | 55.56 |
| | R1-14B | 54.99 | 57.26 | 54.80 | 58.33 | 54.42 | 57.89 | 55.68 | 55.05 |
| | QwQ-32B | 64.65 | 64.89 | 65.03 | 65.28 | 61.87 | 65.21 | 66.16 | 63.32 |
| Avg. | R1-7B | 47.56 | 48.87 | 45.98 | 47.12 | 47.52 | 48.68 | 48.40 | 49.88 |
| | R1-14B | 57.26 | 59.63 | 57.20 | 59.14 | 58.11 | 59.74 | 56.63 | 59.44 |
| | QwQ-32B | 69.99 | 71.75 | 69.15 | 70.96 | 69.71 | 71.02 | 71.44 | 72.00 |

To better evaluate the effectiveness of the routing mechanisms introduced in Section 2.2, we use *Random* routing as an extra baseline. In Table 4, we present the Pass@1 results across four routers on

four benchmarks. It is evident that Top-4 outperforms Top-2 under the *Random* setting, as discussed in Section 3.2. To our surprise, when comparing the results across different routers, we find that *Random* even outperforms *Hybrid*. For example, the average Pass@1 of QwQ-32B reaches 71.75 under the Top-4 setting, which surpasses *Hybrid* by 0.73. This is because in the *Random*, the proportion of dissimilar summaries coming from different peers in the model's sampling paths is higher than that of similar summaries. In contrast, similar and dissimilar summaries each account for half in *Hybrid*. As a result,the *Random* can introduce more diverse or complementary perspectives, providing the model with more useful information. Similarly, under the same setting of Top-4, the average Pass@1 of QwQ-32B for *Dispersed* reaches 72, which surpasses the *Random* by 0.25, as the *Dispersed* can introduce more diverse or complementary perspectives than *Random*.

## E CASE STUDY

### E.1 CASE STUDY ON COMPARING LEAP AND BASELINE WITH BAD BEGINNING

---

**Case Study on LeaP with Bad Beginning**

**Problem:**
Let $A$, $B$, $C$, and $D$ be point on the hyperbola $\frac{x^2}{20} - \frac{y^2}{24} = 1$ such that $ABCD$ is a rhombus whose diagonals intersect at the origin. Find the greatest real number that is less than $BD^2$ for all such rhombi.

**Model Response:** Okay, so I have this problem where there are four points A, B, C, D on the hyperbola given by the equation $\frac{x^2}{20} - \frac{y^2}{24} = 1$. ... So, we need to find the supremum (the least upper bound) of $BD^2 = 4(r^2 + s^2)$, such that the points satisfy the hyperbola equation and the condition $pr + qs = 0$. So, let me formalize this. Given: 1. $\frac{p^2}{20} - \frac{q^2}{24} = 1$. 2. $\frac{r^2}{20} - \frac{s^2}{24} = 1$. 3. $pr + qs = 0$. We need to find the supremum of $4(r^2 + s^2)$. **(Bad Beginning)**
...
<summarize> In conclusion, the current key takeaways and results are: ... Using hyperbola parametrization and vector properties, I derived an expression for BD² in terms of a parameter b, which is tanh u. The expression for BD² is $\frac{80+96b^2}{1-b^2}$ with the constraint $|b| \in (\frac{5}{6}, 1)$. By analyzing the derivative of this function, I found that it's increasing in the interval, implying the minimum BD² occurs as b approaches $\frac{5}{6}$. Therefore, the minimal BD² is achieved in the limit as b approaches $\frac{5}{6}$. </summarize> **(Incorrect Summary)**
<peer_summaries>
Peer 1: "To summarize, based on my previous reasoning, I have currently found that $BD^2$ can be expressed as $80 + (22v)/3$, where $v = s^2$. The variable v is constrained by the equation $u = [600(v - 24)]/(11v - 600)$, where $u = q^2$. The variable v must be either less than 24 or greater than approximately 54.545...
Peer 2: "In short, I've currently concluded that the expression for $BD^2$ in terms of $u = p^2$ is $480 \times \frac{11u-120}{11u-720}$. I also noticed that as $u$ approaches $\frac{720}{11}$ from above, $BD^2$ tends to infinity, which seems contradictory because points on a hyperbola shouldn't allow for infinitely long diagonals...
Peer 3: "To summarize, my recent findings are: I attempted to express $p$ and $s$ in terms of $q$ and $r$ using the condition $pr + qs = 0$, leading to $p = kq$ and $s = -kr$...
Peer 4: "In conclusion, the current insights and results I've gathered are: Expressed BD² in terms of t (where t = q²), transformed the equation into a function f(t) = 80 + [4400(t + 24)] / (11t - 600), and simplified it to f(t) = 480 + (345600/11)/(t - 600/11). The domain of t is t > 600/11. So, as t approaches infinity, f(t) approaches 480, and as t approaches 600/11 from the right, f(t) approaches infinity. Therefore, BD² can be made arbitrarily large, but the problem asks for the greatest real number less than BD² for all such rhombi. So, that number is 480." **(Correct Summary from Peers)** </peer_summaries>
...
From the comments, it appears that some peers have approached the problem differently and arrived at different expressions for BD².Notably, Peer 4 suggests that BD² approaches 480 as t approaches infinity, and that BD² can be made arbitrarily large, but the greatest number less than BD² for all rhombi is 480. **(Self-Verification)** Wait, that seems conflicting with my earlier conclusion where BD² approaches infinity as b approaches 1. So, perhaps I made a mistake in my parametrization. Wait, let me check Peer 4's approach. Peer 4 says: Expressed BD² in terms of t (where t = q²)
...
Therefore, the infimum of BD² is 480, so the greatest real number less than BD² for all rhombi is 480. But in my own reasoning, I thought BD² can be made as large as desired, which would suggest that the infimum is 480, and the supremum is infinity. So, the greatest real number less than BD² for all rhombi is 480...
Therefore, the answer is 480. But wait, in my earlier parametrization, I had BD² tending to infinity as b approaches 1, but here, it's showing that BD² can be made as small as 480 and as large as infinity. Therefore, the greatest real number less than BD² for all rhombi is 480.
So, the answer is 480. **Final Answer** $\boxed{480}$

---

Figure 17: Case Study on LeaP with bad beginning.

> **Case Study on Baseline with Bad Beginning**
>
> **Problem:**
> Let $A$, $B$, $C$, and $D$ be point on the hyperbola $\frac{x^2}{20} - \frac{y^2}{24} = 1$ such that $ABCD$ is a rhombus whose diagonals intersect at the origin. Find the greatest real number that is less than $BD^2$ for all such rhombi.
>
> **Model Response:**
> Okay, so I have this problem where there are four points A, B, C, D on the hyperbola given by the equation $\frac{x^2}{20} - \frac{y^2}{24} = 1$. ... So, we need to find the supremum (the least upper bound) of $BD^2 = 4(r^2 + s^2)$, such that the points satisfy the hyperbola equation and the condition $pr + qs = 0$. So, let me formalize this. Given: 1. $\frac{p^2}{20} - \frac{q^2}{24} = 1$. 2. $\frac{r^2}{20} - \frac{s^2}{24} = 1$. 3. $pr + qs = 0$. We need to find the supremum of $4(r^2 + s^2)$. **(Bad Beginning)**
> ...
> Therefore, f(t) decreases from +infinity to 80 as t increases from 5/6 to infinity. Therefore, the infimum of BD² is 80, but it's never actually reached; it's a limit as t approaches infinity. Therefore, BD² can be made arbitrarily close to 80, but never less than 80. Therefore, the greatest real number less than BD² for all such rhombi is 80. Hence, the answer is $\boxed{80}$.
> **\*\*Final Answer\*\*** $\boxed{80}$

Figure 18: Case Study on baseline with bad beginning.

## E.2 CASE STUDY ON FAILURE EXAMPLES

We provide some case studies to better illustrate our approach. In Figure 19, we present a failure case when applying LeaP to DeepSeek-R1-Distill-Qwen-7B. In this example, the model fails to clearly summarize its own reasoning, which results in the loss of partial reasoning information during summarization. This issue primarily relates to model size. Specifically, when the model is relatively small, incomplete summarization tends to occur more frequently.

In the second case, we demonstrate a situation where the model refuses to engage in communication. As shown in Figure 20, the model assumes it is alone and therefore ignores the perspectives of other peers, continuing its own reasoning trajectory. This case is observed in QwQ-32B but not in the distilled model from R1. We hypothesize that this difference may be related to the training method: QwQ-32B is trained via reinforcement learning (RL), which may lead the model to exhibit higher confidence.

---

**Case Study on Summarization Failure**

**Problem:**
Every morning Aya goes for a 9-kilometer-long walk and stops at a coffee shop afterwards. When she walks at a constant speed of $s$ kilometers per hour, the walk takes her 4 hours, including $t$ minutes spent in the coffee shop. When she walks $s + 2$ kilometers per hour, the walk takes her 2 hours and 24 minutes, including $t$ minutes spent in the coffee shop. Suppose Aya walks at $s + \frac{1}{2}$ kilometers per hour. Find the number of minutes the walk takes her, including the $t$ minutes spent in the coffee shop.

**Model Response:**

...
Wait, wait, from $x^{253} - 34x + 33 = 0$, $x = k^8$, so $x = 1$ is a root, so we can factor it as $(x - 1)(something) = 0$. But since $x = 1$ is a root, and we're interested in the other root, perhaps it's a very small x, but I'm not sure. Wait, perhaps I can write the equation as $x^{253} = 34x - 33$. Hold on, I should summarize the key points briefly to ensure I'm on the right track. <summarize> To summarize, my recent findings are: The problem involves two circles of radius 34 and 2024 arranged in a triangle ABC, with each circle tangent to two sides and the previous circle. The inradius $R$ of ABC can be expressed as $R = 34 \cdot \frac{1-k^8}{1-k}$ for the 34-radius circles, and $R = \frac{1-k^{2024}}{1-k}$ for the 1-radius circles. Setting these equal gives the equation $34 \cdot (1-k^8) = 1-k^{2024}$, which simplifies to $k^{2024} - 34k^8 + 33 = 0$. Letting $x = k^8$, this becomes $x^{253} - 34x + 33 = 0$, which factors as $(x-1)(x^{252} + x^{251} + \cdots + x + 1) - 34x + 33 = 0$. Since $x = 1$ is a root, but we need another real root where $x < 1$. Approximating, $x \approx 0.970588$, leading to $k \approx 0.9963$. Then, $R \approx$ ... (**truncated at 256 tokens**) </summarize>
...

Figure 19: The case study about the reasoning model fail to summarize concisely.

---

**Case Study on Reflection Failure**

**Problem:**
Torus $T$ is the surface produced by revolving a circle with radius 3 around an axis in the plane of the circle that is a distance 6 from the center of the circle (so like a donut). Let $S$ be a sphere with a radius 11. When $T$ rests on the inside of $S$, it is internally tangent to $S$ along a circle with radius $r_i$, and when $T$ rests on the outside of $S$, it is externally tangent to $S$ along a circle with radius $r_o$. The difference $r_i - r_o$ can be written as $\frac{m}{n}$, where $m$ and $n$ are relatively prime positive integers. Find $m + n$. [asy] unitsize(0.3 inch); draw(ellipse((0,0), 3, 1.75)); draw((-1.2,0.1)..(-0.8,-0.03)..(-0.4,-0.11)..(0,-0.15)..(0.4,-0.11)..(0.8,-0.03)..(1.2,0.1)); draw((-1,0.04)..(-0.5,0.12)..(0,0.16)..(0.5,0.12)..(1,0.04)); draw((0,2.4)--(0,-0.15)); draw((0,-0.15)--(0,-1.75), dashed); draw((0,-1.75)--(0,-2.25)); draw(ellipse((2,0), 1, 0.9)); draw((2.03,-0.02)--(2.9,-0.4)); [/asy]

**Model Response:**
...
<peer_summaries>
...
Peer 4: "In conclusion, the current insights and results I've gathered are: By setting up the equations for the sphere and the torus, and analyzing the gradients for tangency, I derived that at points of tangency, the z-component gives $k = 2$ (assuming $z \neq 0$). This led to an equation relating the radial distance $\rho = sqrt(x^2 + y^2)$ to the major radius $R = 6$. Solving this gives a specific $\rho$ value, which when substituted back into the equations, yields a valid solution for $z$. This allows computing the radii $r_i$ and $r_o$ for internal and external tangency, leading to the desired difference. However, I must ensure that the configuration (internal vs. external) corresponds correctly to the direction of the normals." </peer_summaries> Hmm, it seems that my peers have given me some comments, so let me check if anyone's conclusions are different from mine before I continue my own reasoning. But since I'm alone, I'll proceed. Let me try to proceed with the standard equation of the torus. Let me define the torus such that its $axis$ is the $x-axis$. The generating circle has a major radius $R = 6$, so its center is at $(6, 0, 0)$, and minor radius $r = 3$. The equation of the torus is: $(sqrt(y^2 + z^2))^2 + (x-6)^2 = 3^2$ ? Wait, no, perhaps I need to think differently. Let me recall that the standard equation for a torus rotated around the $x-axis$ with major radius R and minor radius r is: $(sqrt((sqrt(y^2 + z^2) - R)^2 + x^2))$? Wait, perhaps I should use a different approach.
...

Figure 20: The case study that the QwQ-32B reject to understand the summaries from peers.

### E.3 CASE STUDY ON SUCCESS EXAMPLES

In the third case, we present a scenario where the model ultimately reaches the correct answer through interaction with its peers. As shown in Figure 21, the model initially fails to fully summarize its own reasoning process, leading to the loss of partial reasoning information during summarization. However, during the subsequent interaction with peers, the model receives a clearer and more explicit reasoning path provided by them. By verifying the summaries generated by other peers, the model gradually identifies the correct solution to the problem.

In the fourth case, we present a scenario where the model initially generates an incorrect answer during the reasoning process. Meanwhile, other peers also produce flawed summaries, though some provide correct ones. Through interaction and mutual verification, the model eventually identifies the correct reasoning path. As shown in Figure 22, the model first summarizes its own reasoning and arrives at the incorrect answer of 389. Peer 1 and Peer 2 provide incorrect summaries, while only Peer 3 offers a reasonable one. By verifying the summaries from all peers, the model determines that the summaries from Peer 1 and Peer 2 are inconsistent. In contrast, the summary from Peer 3 is considered reliable. Based on this verification process, the model gradually uncovers the correct solution to the problem.

> **Case Study on LeaP Success**
>
> **Problem:**
> "Each vertex of a regular octagon is independently colored either red or blue with equal probability. The probability that the octagon can then be rotated so that all of the blue vertices end up at positions where there were originally red vertices is $\frac{m}{n}$, where $m$ and $n$ are relatively prime positive integers. What is $m + n$?"
>
> **Model Response:**
>
> ...
> <peer_summaries>
>
> ...
> <summarize> In conclusion, the current insights and results I've gathered are as follows: - **Good Subsets**: Subsets with 0 to 3 elements are all good, totaling $1 + 8 + 28 + 56 = 93$ subsets. Subsets of size 4 may include some bad subsets, while subsets of size 5–8 are mostly bad, though not all. - **Bad Subsets of Size 4**: Examples like 0,1,3,4 cannot be rotated to be disjoint, indicating that some size-4 subsets are bad. Calculating their exact count requires identifying patterns or symmetries that make these subsets resistant to rotation. - **Symmetry and Complementarity**: The problem's symmetry suggests that the number of bad subsets of size $k$ equals those of size $8 - k$. However, since subsets of size 3 (complement of 5) are all good, subsets of size 5 must also be good, contradicting earlier assumptions.This implies a need to reevaluate. - **Reevaluation**: Perhaps subsets of size 5 are actually good, and the real bad subsets are only those of size 7, 8, and some specific size 4 subsets. This would mean the total bad subsets are 1 (... </summarize> (**truncated at 256 tokens**) <peer_summaries> Peer 1: "..."Peer 2:"..."Peer 3: "..."Peer 4:"To summarize, my recent findings are as follows: - **Subsets of Size 4**: - **Rotation by 1 or 7**: Only two valid subsets (checkerboard patterns 0,2,4,6 and 1,3,5,7). - **Rotation by 2 or 6**: Requires selecting opposite pairs in each of the two 4-cycles formed by the rotation. This yields $2 \times 2 = 4$ subsets. - **Rotation by 4**: Selecting one vertex from each of the four diametrical pairs, resulting in $2^4 = 16$ subsets. - **Overlap Considerations**: No overlaps between these categories, as the subsets from different rotations are structurally distinct. - **Total Valid Size 4 Subsets**: $2 + 4 + 16 = 22$. - **Total Favorable Subsets**: - Size 0:1 - Size 1:8 - Size 2:28 - Size 3:56 - Size 4:22 - **Total**:$1+8+28+56+22 = 115$ - **Probability Calculation**: - Total subsets:..."</peer_summaries> Hmm, it seems that my peers have given me some comments, so let me check if anyone's conclusions are different from mine before I continue my own reasoning. Peer 1 is talking about independent sets on a cycle with distance constraints. Peer 2 suggests that size4 subsets can be counted by analyzing different rotation types, leading to 22 valid subsets. Peer 3 mentions inclusion-exclusion over rotation classes. Peer 4 calculates total favorable subsets as 115, leading to probability 115/256. Wait, Peer4's calculation: Size0:1 Size1:8 Size2:28 Size3:56 Size4:22 (only those that are good) Total: $1+8+28+56+22 = 115$ Thus probability 115/256, which simplifies to 115 and 256 are coprime? 256 is $2^8$. 115 is $5 \times 23$, so yes. Thus m+n=115+256=371. But I need to verify if this is correct. According to Peer4's reasoning: For size4 subsets, the valid (good) subsets are 22. The bad subsets of size4 are 70-22=48. Therefore, total bad subsets are: Size5:56 Size6:28 Size7:8 Size8:1 Size4 bad:48 Total bad:56+28+8+1+48=141 Thus total good subsets:256-141=115 Therefore, probability 115/256 Thus the answer is 115+256=371. But I need to verify if Peer4's count of size4 subsets is correct. Peer4 says
> ...
> Yes. Thus, these subsets are good. The checkerboard patterns are also good. Therefore, the count of22 valid subsets of size4 is correct. Thus, the total good subsets are115, leading to the probability of115/256. Thus, the final answer is115 +256= 371.( **Final Answer** $\boxed{371}$ )

Figure 21: The case study that the model learn from peers successfully, which lead to a correct final answer.

> **Case Study on LeaP Success**
>
> **Problem:**
> "Each vertex of a regular octagon is independently colored either red or blue with equal probability. The probability that the octagon can then be rotated so that all of the blue vertices end up at positions where there were originally red vertices is $\frac{m}{n}$, where $m$ and $n$ are relatively prime positive integers. What is $m + n$?"
>
> **Model Response:**
>
> ...
>
> <summarize> In conclusion, the current insights and results I've gathered are as follows: - The problem involves two configurations of circles arranged in triangle ABC: eight circles of radius 34 and 2024 circles of radius 1,
>
> ...
>
> - After extensive exploration and considering peer insights, the most consistent approach involves setting up equations based on the path lengths required by both configurations and solving for the inradius using proportional reasoning, leading to an inradius of $(\frac{384}{5})$, ( **resulting in** $(m + n = 384 + 5 = 389...$ )</summarize> <peer_summaries>Peer 1: "To summarize, based on my previous reasoning, I have currently found
>
> ...
>
> Thus, GCD is 1, making $\frac{68,800}{2007}$ already in simplest terms. Adding numerator and denominator gives $68,800 + 2007 = 70,807$. However, this contrad..." Peer 2: "In short, my current conclusions are that the problem involves arranging circles in geometric progression along an angle bisector of triangle ABC's vertex B.
>
> ...
>
> The relationship $2024 = 8 \times 253$ hints that $k^{2016} = 34$ simplifies to $(k^8)^{252} = 34$, implying $k^8 = \sqrt[252]{34}$. This still leaves the inradius..." Peer 3: "In short, I've currently concluded that by modeling the problem using a geometric progression of radii along the angle bisector at vertex B, and setting up equations for both configurations (8 circles of radius 34 and 2024 circles of radius 1), I derived that the sine of half the angle $\theta/2$ is $\frac{11}{1190}$. Substituting back into the inradius formula yields $R = \frac{192}{5}$, which simplifies to $\frac{192}{5}$. Since 192 and 5 are coprime, the final answer is $m + n = 197$. This result aligns with the calculations when assuming the incenter is positioned such that the distance from vertex B to the incenter along the angle bisector equals the distance to the last circle plus its radius." Peer 4: "In conclusion, the current key takeaways and results are as follows: By modeling the problem using geometric progressions for the radii of the circles arranged along the angle at vertex B,
>
> ...
>
> the expression for $R$ simplifies to $R = \frac{2 \times 34^{253/252}}{34^{1/2016} + 1}$. Recognizing that $34^{253/252} = 34^{1+1/252}$, this becomes R =..." </peer_summaries>
>
> ...
>
> Peer 3 suggests an inradius of 192/5, so 192+5=197. Peer 1 and 2 are getting large fractions. Peer 4's answer is unclear. Given that in the Asymptote example, the inradius was about 0.45 for small circles, and the problem has 2024 circles of radius 1,
>
> ...
>
> which is not a fraction. Alternatively, the answer is 34/253 +1/2024? No. Alternatively, the problem is a trick question where the inradius is simply the sum of the radii multiplied by the number of circles divided by some number. Alternatively, the inradius is the solution to the equation: 8*(2*34) =2024*(2*1) Which is 544 =4048 $\rightarrow$ impossible. Thus, the only plausible answer is Peer3's 192/5, which is 38.4.
>
> ...
>
> Given that 384/5 is 76.8, and the path length for first case is 544, which is 7*68 + 68=544, and 544/7=77.714. Close to 76.8. Perhaps Peer1's calculation is correct. Alternatively, the inradius is 544/7=77.714= 544/7=77.714285..., which is 544 divided by7 is 77.714. 544 divided by7:7×77=539, so 544/7=77+5/7=77.714285... Thus, as a fraction, 544/7, which reduces to 544/7 (since 544 and7 are coprime). Thus, m+n=544+7=551.
>
> . . .
>
> then r=34 +476 sin$\theta$ To also satisfy the second configuration: r=1 +4046 sin$\theta$ Thus, 34+476 sin$\theta$=1+4046 sin$\theta$ 33= 3570 sin$\theta$ sin$\theta$=33/3570=11/1190=0.0092 Thus, sin$\theta$=0.0092 Then, r=1 +4046*(11/1190)=1 + (4046×11)/1190 Calculate numerator:4046×11=44506 Divide by1190: 44506/1190 =37.4 Thus, r =1+37.4=38.4=384/10=192/5=38.4 Ah, so this gives r=192/5=38.4 Thus, m=192, n=5, so m+n=197 This matches Peer3's answer. Therefore, the inradius is 192/5. The steps: Assuming the progression ratio k=1 (even though impossible), then the inradius is derived as above, leading to sin$\theta$=11/1190, and substituting back gives r=192/5. **Thus, the answer is 192+5=197. \*\*Final Answer\*\*** $\boxed{197}$ ) ...

Figure 22: The case study that the model learn from peers successfully, which lead to a correct final answer.

## F    TRAINING DETAILS OF LEAP-T

We utilize 8 NVIDIA A100 GPUs to train and evaluate our models. We use LLama-Factory Zheng et al. (2024) to train our models on AIME 1984 to 2023. [1] The training hyperparameters are as follows:

Table 5: Hyperparameters for LeaP-T training.

| Hyperparameter | 1.5B Model | 7B Model | 14B Model |
|---|---|---|---|
| Batch size | 16 | 16 | 16 |
| Learning rate | $1 \times 10^{-5}$ | $1 \times 10^{-5}$ | $1 \times 10^{-5}$ |
| Learning rate scheduler | Cosine decay | Cosine decay | Cosine decay |
| Warmup ratio | 0.05 | 0.05 | 0.05 |
| Optimizer | AdamW | AdamW | AdamW |
| Weight decay | $1 \times 10^{-4}$ | $1 \times 10^{-4}$ | $1 \times 10^{-4}$ |
| Max sequence length | 16K | 16K | 16K |
| Training epochs | 8.0 | 8.0 | 5.0 |
| Precision | bfloat16 | bfloat16 | bfloat16 |

## G    COMPARISON WITH MoA

Table 6: Performance vs. compute cost comparison under 7B scale.

| Method | Score (%) | Compute (Avg. Tokens per Question) |
|---|---|---|
| R1-7B | 66.67 | ∼74,000 |
| MoA + R1-7B | 53.33 | ∼77,975 |
| LeaP + R1-7B | 71.67 | ∼69,860 |
| LeaP-T 7B | 73.33 | ∼67,998 |

We also test MoA Wang et al. (2024) as a baseline under the setting of 4 Layers and 3 Agents, with the prompts available in Appendix B. As shown in Table 6, MoA costs approximately 80K tokens per problem, but the results show that it does not transfer well to LRMs. The reason is that LRMs cannot follow the user's instructions effectively. We find that even when the correct answer is achieved in the intermediate layers, the model still reaches incorrect conclusions in the final layer.

## H    EVALUATION RESULTS OF LEAP IN SINGLE COMMUNICATION ON FOUR REASONING BENCHMARKS

We evaluate a simplified LeaP variant (LeaP-S) that retains only the first LeaP block at $T$ tokens and continues generation along a single path. This mirrors how humans often work: gathering ideas early, then independently verifying them. As shown in Table 7, this single communication variant outperforms DeepSeek-R1-Distill-Qwen-14B. For example, on AIME 2024, it improves Pass@1 from 64.47 to 68.13; on GPQA Diamond, from 53.47 to 57.42. Even with just one block, it achieves a higher average Pass@1 (55.95 vs. 53.86), showing that early peer exposure effectively guides reasoning. Gains are most notable in math and multi-hop tasks, where early external signals reduce error accumulation. This variant also balances performance and efficiency, making it suitable for single-path scenarios. We further explore different hyperparameters on AIME 2024 (Table 8). Increasing the interval $T$ from 2K to 4K improves accuracy (e.g., 67.92 to 69.48 at $k = 8$), and using more peers (e.g., $k = 8$ vs. $k = 2$ at $T = 4$K) also yields better results.

---

[1]LLama-Factory is licensed under the Apache-2.0 license. All data from AIME is licensed under the MIT license.

Table 7: The Pass@1 results of independent reasoning and LeaP-S on AIME 2024, AIME 2025, AIMO 2025 and GPQA Diamond for 14B model.

| Benchmarks | R1-14B | $T = 2\mathbf{K}$ | $T = 4\mathbf{K}$ |
|---|---|---|---|
| AIME 2024 | 64.47 | 66.25 | **68.13** |
| AIMO 2025 | 46.87 | 49.38 | **50.94** |
| AIME 2025 | **48.64** | 45.83 | 47.29 |
| GPQA Diamond | 53.47 | 55.11 | **57.42** |
| Avg. | 53.86 | 54.14 | **55.95** |

Table 8: The Pass@1 results for varying $k$ values and position of LeaP block on AIME 2024 with the 14B model in LeaP-S.

| | $k = 2$ | $k = 4$ | $k = 6$ | $k = 8$ | $k = 16$ (Top-2) | $k = 16$ (Top-4) |
|---|---|---|---|---|---|---|
| $T = 2k$ | 65.21 | 66.25 | 65.73 | 67.92 | 66.46 | 66.86 |
| $T = 4k$ | 65.73 | 68.13 | 68.85 | 69.48 | 67.92 | 69.27 |

To contextualize the computational cost of this approach, we analyze its efficiency. In the LeaP-S experiment with $N = 4$ paths and communication at $T = 4K$ tokens, the generation of three peer paths creates an initial overhead of $12,288$ tokens. Due to the quadratic complexity ($O(L^2)$) of the attention mechanism with respect to sequence length, this initial investment results in an approximate 50% increase in computational cost (FLOPs) compared to generating a single path. To verify this is an efficient use of compute, we provide a comparison against self-consistency baselines (`Cons@2` and `Cons@3`), which represent alternative ways to spend a similar or larger computational budget. As shown in Table 9, LeaP-S significantly outperforms `Cons@3` (68.13 vs. 66.67) while consuming fewer tokens overall. This demonstrates that the early, targeted communication in LeaP-S is a more compute-efficient strategy for improving reasoning than generating additional independent solutions for voting.

Table 9: Comparison of LeaP-S with independent reasoning (`Cons@k`) baselines on AIME 2024. LeaP-S achieves a higher score with fewer tokens than `Cons@3`, demonstrating superior computational efficiency.

| Method | Score | Tokens per Question |
|---|---|---|
| R1-14B + LeaP-S* ($N = 4$) | **68.13** | 23,252 |
| R1-14B `Cons@2` | 53.33 | 20,944 |
| R1-14B `Cons@3` | 66.67 | 29,319 |

# I    OTHER IN-DEPTH ANALYSIS

## I.1    ON COMPUTATIONAL OVERHEAD

We next analyze the computational overhead of LeaP. The additional cost comes from processing peer summaries at communication turns. Importantly, while these summaries add tokens to the context, they are processed in the parallelized prefill stage, whereas the dominant latency factor is the serial autoregressive generation.

### I.1.1    FLOPS

To analyze the FLOPs overhead, we first define the key architectural parameters and derive the FLOPs for a standard forward pass.

The FLOPs for processing a sequence of length $N$ can be broken down as follows.

**Per-layer FFN FLOPs.**
$$FLOPs_{\text{FFN}} = 4 \times N \times d_{model} \times d_{ff} \qquad (5)$$

| Symbol | Definition |
|---|---|
| $d_{model}$ | Model hidden dimension (hidden_size) |
| $d_{ff}$ | Feed-forward intermediate dimension |
| $n_{layers}$ | Total number of layers |
| $n_{heads}$ | Number of attention heads for Query |
| $n_{kv\_heads}$ | Number of attention heads for Key/Value (GQA) |
| $d_{head}$ | Dimension of each attention head |
| $v_{size}$ | Vocabulary size |
| $N$ | Input sequence length |

Table 10: Notation for FLOPs analysis.

**Per-layer Self-Attention FLOPs.**

$$FLOPs_{\text{Attn}} = 2Nd_{model}^2 + 4Nd_{model}n_{kv\_heads}d_{head} + 2N^2 d_{model} \tag{6}$$

$$+ 2N^2 d_{model} + 2Nd_{model}^2 \tag{7}$$

$$= 4N^2 d_{model} + 4Nd_{model}^2 + 4Nd_{model}n_{kv\_heads}d_{head} \tag{8}$$

**Total FLOPs.**

$$FLOPs(N) = n_{layers} \times \left( FLOPs_{\text{FFN}} + FLOPs_{\text{Attn}} \right) + 2Nd_{model}v_{size} \tag{9}$$

$$= n_{layers} \times \left( 4N^2 d_{model} + 4Nd_{model}(d_{ff} + d_{model} + n_{kv\_heads}d_{head}) \right) \tag{10}$$

$$+ 2Nd_{model}v_{size} \tag{11}$$

For simplicity we approximate the dominant terms as:

$$FLOPs(N) \approx 4 \cdot n_{layers} \cdot N \cdot d_{model} \cdot (N + d_{ff} + d_{model}) \tag{12}$$

The extra FLOPs from LeaP occur during communication turns. At each turn $i$ (which occurs at a context length of $i \cdot T$), the model processes new summary tokens. The cost of adding $L_{\text{new}} = (k+1) \times L_{\text{sum}}$ tokens to a context of length $M$ is:

$$\Delta FLOPs(M, L_{\text{new}}) = FLOPs(M + L_{\text{new}}) - FLOPs(M) \tag{13}$$

Example: For R1-7B with $T = 4096$, $k = 4$, $L_{\text{sum}} = 143$ ($L_{\text{new}} \approx 715$), and an average of $n_{\text{turns}} = 1.41$ communication rounds:

$$FLOPs_{\text{Overhead}} \approx \left[ FLOPs(4096 + 715) - FLOPs(4096) \right] \tag{14}$$

$$+ 0.41 \times \left[ FLOPs(8192 + 2 \times 715) - FLOPs(8192 + 715) \right] \tag{15}$$

The baseline generates on average $9,045$ tokens:

$$FLOPs_{\text{Baseline}} \approx FLOPs(9045) \tag{16}$$

Therefore, the relative overhead is:

$$\text{Overhead (\%)} = \frac{FLOPs_{\text{Overhead}}}{FLOPs_{\text{Baseline}}} \times 100\% \approx 12.08\% \tag{17}$$

This confirms that while LeaP introduces a measurable FLOPs cost at each communication step, this overhead is moderate and is offset by the shorter reasoning paths it enables, thereby reducing the total number of decode steps.

### I.1.2 TOTAL TOKENS

We analyze LeaP's inference efficiency based on the total number of tokens and the frequency of self-correction, often termed "aha" moments. As illustrated in Figure 23, LeaP does not generate significantly more tokens than the baseline across the QwQ-32B, DeepSeek-R1-Distill-Qwen-7B, and DeepSeek-R1-Distill-Qwen-14B models. In fact, for certain benchmarks like GPQA Diamond, its average response length is sometimes shorter.

We further analyze "aha" moments using keywords from previous studies (Zeng et al., 2025a; Li et al., 2025; Chen et al., 2025). The results consistently show that LeaP exhibits fewer "aha" moments than the baseline. For example, with QwQ-32B, LeaP has 16.4% fewer such instances across three math benchmarks. Additionally, we observe that the number of "aha" moments for Top-4 settings is consistently lower than for Top-2 settings. These findings indicate that receiving peer opinions and results reduces the model's need for reflection and self-correction, leading to a more streamlined reasoning process.

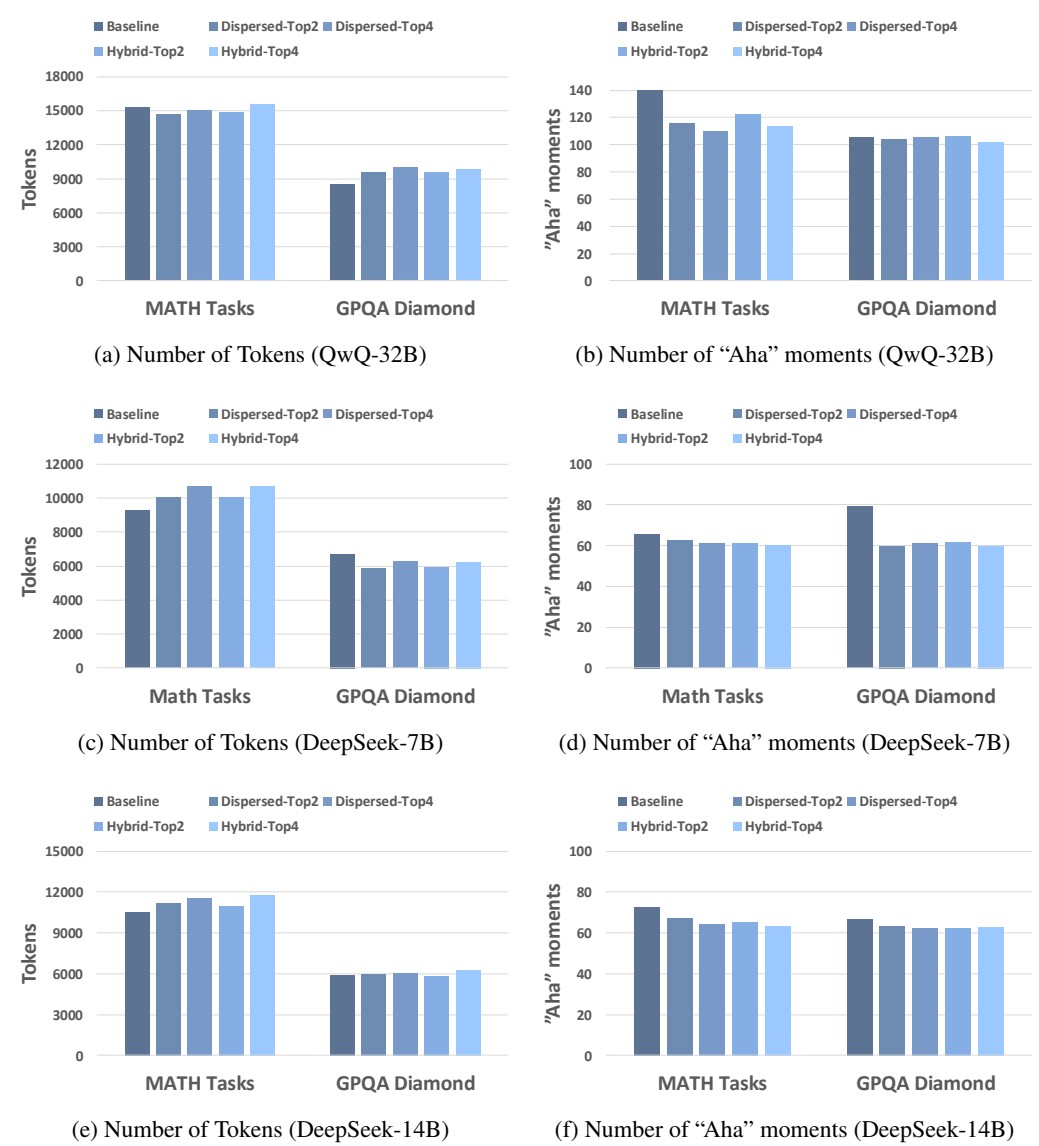

(a) Number of Tokens (QwQ-32B)     (b) Number of "Aha" moments (QwQ-32B)

(c) Number of Tokens (DeepSeek-7B)     (d) Number of "Aha" moments (DeepSeek-7B)

(e) Number of Tokens (DeepSeek-14B)     (f) Number of "Aha" moments (DeepSeek-14B)

Figure 23: We illustrate the average number of tokens and "Aha" moments on QwQ-32B (top), DeepSeek-R1-Distill-Qwen-7B (middle), and DeepSeek-R1-Distill-Qwen-14B (bottom). Our method produces a comparable number of tokens to the baseline, while yielding fewer "Aha" moments across all models.

## I.2 ON HOMOGENIZATION

We acknowledge that homogenization is a factor in designing the LeaP framework, our analysis in Section 5.1.3 and Section 5.1.4 shows that earlier communication between peers leads to more similar results. However, as the main experiments indicate, the overall benefit of communication, especially when it doesn't occur too early, significantly outweighs the risk of this convergence.

An insightful hypothesis is that the diversity in our summary prompting format (as shown in Figure 13) helps mitigate this homogenization. To test this, we conducted an ablation comparing our default setting (random prompts from Figure 13) with a fixed trigger and template. The fixed prompt was:

> **Summary Trigger:** *Alright, let's take a step back and summarize what we've figured out so far.*
> **Summary Template:** *In short, my current conclusions are that*

Table 11: Ablation study on the impact of prompt diversity on homogenization. Performance differences are minimal, suggesting prompt diversity is not the primary factor.

| Pass@1 (R1-7B) | Random Prompt | Fixed Prompt |
|---|---|---|
| AIME 2024 | 60.52 | 60.21 |
| AIME 2025 | 38.44 | 40.48 |
| AIMO 2025 | 45.00 | 40.31 |

The results in Table 11 show only minor performance differences between using random and fixed prompts. This suggests that while prompt diversity may contribute slightly, it is **not the decisive factor** in preventing homogenization or in LeaP's overall success. This finding reinforces our main conclusion: **the core benefit comes from the peer-learning mechanism itself**, and this benefit significantly outweighs the risk of convergence, especially when communication is timed appropriately.

## I.3 ON DESIGN OF ROUTERS

### I.3.1 NECESSITY OF THE ROUTER (ROUTER VS. BROADCAST)

To validate the necessity of a selective routing mechanism, we conducted an ablation study comparing our proposed router against a "broadcast" baseline. In the broadcast setting, every reasoning path receives summaries from all other paths, which is equivalent to setting the number of peers $k$ to the total number of paths ($k = 32$ in our experiments). This setup effectively disables selective communication. All other experimental parameters were held constant to ensure a fair comparison.

The results, presented in Table 12, show that our selective routing approach (Top-4) consistently outperforms the full broadcast baseline across all benchmarks. This finding suggests that merely increasing the volume of information is suboptimal and can be detrimental to performance.

Table 12: Comparison of Pass@1 performance between our selective router ($k = 4$) and a broadcast baseline ($k = 32$). Selective routing consistently yields better results.

| Benchmark | Pass@1 (w/ Router, $k = 4$) | Pass@1 (Broadcast, $k = 32$) |
|---|---|---|
| AIME 2024 | 60.52 | 59.03 |
| AIME 2025 | 38.44 | 37.71 |
| AIMO 2025 | 45.00 | 41.88 |

This conclusion is further corroborated by our analysis of communication traffic in Section 5.1.2 (Figure 7), where we demonstrate that performance does not increase monotonically with the number of peer summaries ($k$). Instead, a clear trade-off emerges:

- **Too few summaries** ($k < 4$) provide insufficient information for robust self-correction and refinement.

- **Too many summaries** ($k > 4$) can introduce distracting or contradictory information, overwhelming the model's reasoning capabilities and leading to cognitive overload.

Our experiments identify $k = 4$ as the optimal configuration, striking a balance between providing rich, diverse insights and avoiding information overload.

### I.3.2 SEMANTIC-BASED ROUTER

To explore more advanced routing strategies, we conducted an experiment using semantic similarity as the basis for routing, rather than the heuristic-based Levenshtein distance. For this experiment, we employed the `bge-large-en-v1.5` (Xiao et al., 2024) model to compute embeddings for each summary. We then applied the same Dispersed Routing strategy, selecting the Top-4 most dissimilar summaries based on cosine similarity. All other settings were consistent with those described in Section 2.2.

Table 13: Performance comparison between the heuristic-based (Levenshtein) router and a semantic-based (embedding similarity) router. The semantic approach shows modest but consistent improvements.

| Benchmark | Heuristic-based | Semantic-based |
|---|---|---|
| AIME 2024 | 60.52 | 61.56 |
| AIME 2025 | 38.44 | 39.48 |
| AIMO 2025 | 45.00 | 45.21 |

As shown in Table 13, routing based on semantic dissimilarity yields a consistent, albeit modest, performance improvement. This result validates the intuition that a deeper, meaning-based understanding of peer summaries can lead to more effective information exchange. However, this approach introduces the computational overhead of encoding the summaries. We believe this points toward a promising future direction where peer-learning paradigms are integrated directly into the model architecture. Such an integration could enable routing based on internal feature similarity, which might prove to be both more effective and more computationally efficient than methods relying on explicit natural language summaries.

### I.4 OTHER SENSITIVITY ANALYSIS

### I.4.1 ON TEMPERATURE

We conduct an in-depth analysis of the temperature parameter $\tau$ by varying it from 0.1 to 1.0, while keeping all other settings fixed. We use Top-4 Dispersed routing and set the communication interval to $T = 4k$ tokens. As shown in Figure 24, when the temperature is low ($\tau \leq 0.3$), the model achieves an average accuracy of around 71, similar to the *Clustered* router baseline. This is expected, as a low temperature reduces output diversity, thereby limiting the benefits of peer communication. On the other hand, when the temperature is too high (e.g., $\tau \geq 0.9$), performance drops noticeably. This is likely due to excessive randomness, which may lead the model away from coherent reasoning or instruction following. Overall, a moderate temperature appears to strike a good balance between diversity and stability.

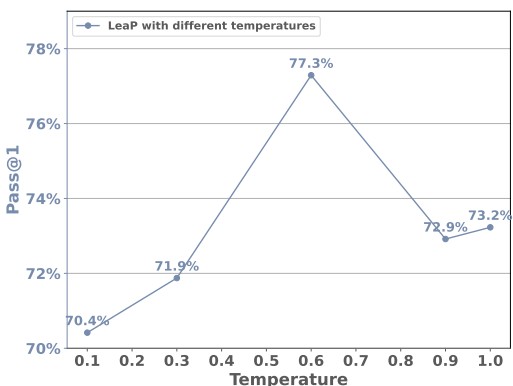

Figure 24: Pass@1 results of various temperatures on AIME 2024.

### I.4.2 ON COMMUNICATION GRANULARITY $T$ FOR LEAP-T

To further study the effect of communication granularity, we analyze LeaP-T under different token intervals $T$. As shown in Figure 25a, for the 7B version, the Pass@1 score on AIME 2024 decreases from 66.35 to 61.25 as $T$ increases, while the number of generated tokens drops accordingly. This trend is consistent with the analysis in Section 5.1.1: more frequent communication (i.e., smaller $T$) improves performance slightly, but increases token consumption due to more frequent summarization and message exchange. We observe a similar trend for the 14B version in Figure 25b.

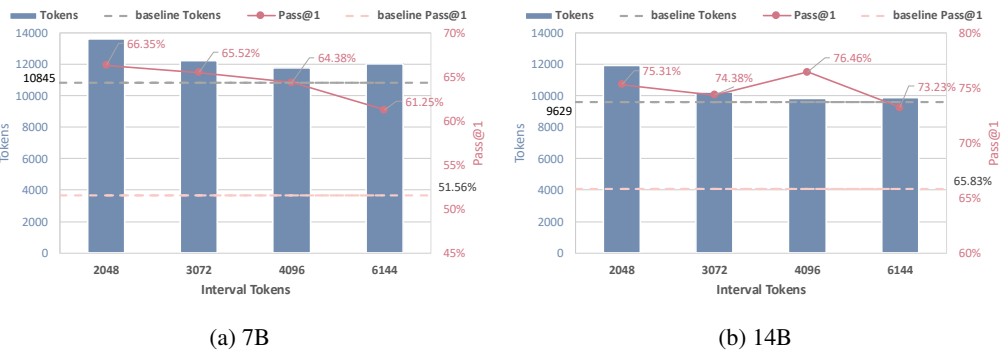

|          |          |
|:--------:|:--------:|
| (a) 7B   | (b) 14B  |

Figure 25: Pass@1 and total tokens on AIME 2024 for LeaP-T-7B and LeaP-T-14B models, evaluated across interval tokens of LeaP from 2048 to 6144.

### I.4.3 ON COMMUNICATION POSITION FOR LEAP-T

Furthermore, Figure 26 analyzes when communication is most effective. The results indicate that performance peaks when the LeaP block is inserted at 4K tokens. Specifically, Pass@1 increases from 68.85 at 2K to 71.77, then declines to 69.69 at later positions. Although a slight recovery to 71.25 is observed, the overall pattern suggests that earlier communication tends to yield better results. These results are consistent with our analysis in Section 5.1.4, and highlight the importance of timely information exchange in improving performance.

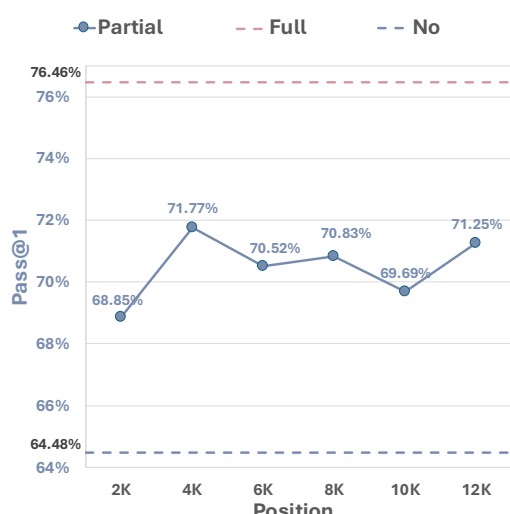

Figure 26: Performance of partial LeaP on the 14B model with LeaP-T, evaluated across the positions of LeaP block ranging from 2K to 12K tokens

## I.5 ROBUSTNESS AT VARIOUS DIFFICULTY LEVELS FOR LEAP-T

We also report the performance of LeaP-T across the five difficulty levels of AIME 2024. These difficulty levels follow the same categorization introduced in Section 5.2.2, based on model (without peer reasoning) accuracy. To exclude the impact of distilling with a 32B model, we conduct distillation using SFT without LeaP, which serves as our baseline. This setup allows for a direct comparison with our LeaP-T approach. Across all five difficulty levels, LeaP-T generally outperforms the baseline. For instance, as shown in Figure 29, LeaP-T achieves higher accuracy while consuming fewer tokens. These findings are consistent with the analysis presented in Section 5.2.2. This is because LeaP facilitates earlier consensus during the reasoning process, thereby reducing unnecessary computational overhead caused by overthinking. Notably, the improvement of LeaP-T over the baseline suggests that its performance gain is not solely attributed to knowledge transfer through distillation. Instead, it underscores the effectiveness of a training paradigm centered on *learning from peers*.

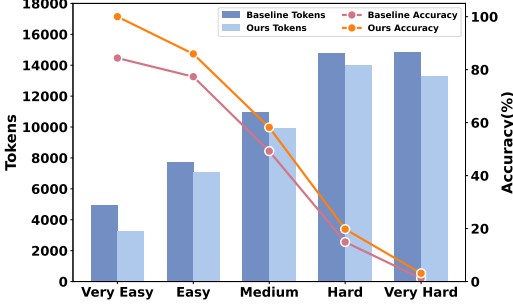

Figure 27: The Pass@1 and average token distribution across different difficulty levels, from Very Easy to Very Hard, for the LeaP-T-1.5B.

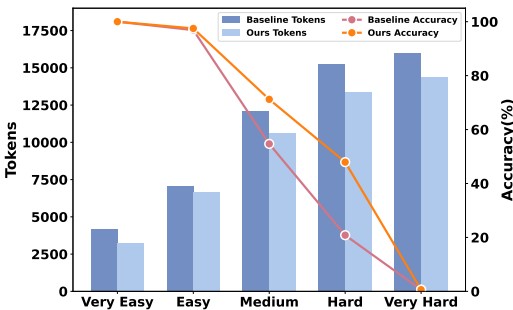

Figure 28: The Pass@1 and average token distribution across different difficulty levels, from Very Easy to Very Hard, for the LeaP-T-7B.

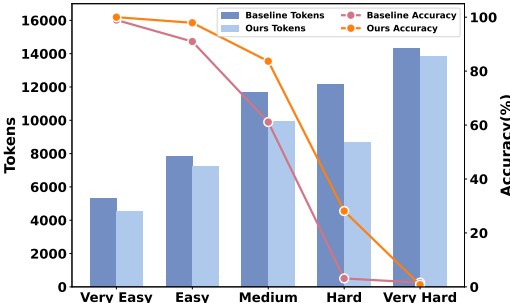

Figure 29: The Pass@1 and average token distribution across different difficulty levels, from Very Easy to Very Hard, for the LeaP-T-14B.

### I.6 BEHAVIOR DIFFERENCE OF LEAP USING RL-*ed* AND NON-RL MODELS

We observe that reinforcement learning (RL) models, such as QwQ, display higher confidence in their reasoning. This is particularly evident in the increasing proportion of *Unaffected* cases at later stages of reasoning. The reinforcement learning process tends to encourage more consistent behavior, with certain tokens' sampling probabilities increasing, guiding the model back to familiar reasoning paths. This aligns with findings from previous studies Ji et al. (2025); Yu et al. (2025), which show that RL-trained models tend to prefer their learned reasoning strategies, exhibiting greater self-assurance in their conclusions. This confidence may explain the model's resistance to altering its reasoning path when provided with peer insights, especially in the later stages of reasoning.

## J LIMITATIONS AND FUTURE WORK

While LeaP shows strong empirical success, we identify several limitations and directions for future refinement. First, the current **routing mechanisms** relies on heuristic similarity (e.g., Levenshtein distance), which may not fully capture semantic nuances; more sophisticated relevance metrics could be beneficial. Second, LeaP's effectiveness currently depends on **high-quality peer summaries** communicated via explicit natural language, which incurs token overhead. A promising future direction is to move the communication mechanism inside the model architecture itself, enabling interaction directly in the **latent space**. This model-level change would eliminate token overhead entirely, broaden the applicability of peer-learning, and reduce the reliance on explicit summary quality.

## K THE USE OF LARGE LANGUAGE MODELS (LLMS)

In this work, we utilized Large Language Models (LLMs), such as ChatGPT and Google Gemini, solely for the purpose of polishing and improving the readability of our manuscript. The core ideas, experimental results, and analyses were exclusively conducted by the authors.

