# OpenReview forum: "Learning from Peers in Reasoning Models"
_ICLR.cc/2026/Conference — ICLR 2026 Conference Withdrawn Submission_

### Official Review · Reviewer_46eR · 2025-10-29

**Soundness:** 2
**Presentation:** 2
**Contribution:** 2
**Rating:** 2
**Confidence:** 3

**Summary:**

The paper “Learning from Peers in Reasoning Models (LeaP)” introduces a collaborative reasoning framework for large language models (LLMs) that enables multiple reasoning paths to exchange information during inference. The authors show a “Prefix Dominance Trap,” where an early flawed reasoning prefix can significantly reduce accuracy, and propose LeaP as a solution. In this method, several reasoning paths periodically summarize their intermediate thoughts and share them through a selective routing process, allowing each path to revise its reasoning based on peer input. Experiments across math and reasoning benchmarks, including AIME and GPQA, show that this approach improves accuracy and robustness across models of different sizes, even when some paths start from weak reasoning prefixes

**Strengths:**

- The topic is timely and relevant, aligning with current research trends in parallel reasoning
- The paper is clearly written and easy to follow
- The core idea—allowing reasoning paths to communicate and cross-correct—is intuitive and conceptually appealing

**Weaknesses:**

I don’t find the empirical comparison entirely fair.
- Table 1: It presents Pass@1 for independent reasoning, self-correct prompt, and LeaP. However, each LeaP path benefits from multiple peer paths, so there is no well-defined Pass@1 for LeaP. A more reasonable evaluation, in my view, would be: assuming the width of each LeaP inference is N, run M LeaP inferences and M×N independent inferences, and then report Pass@N for a fair comparison.
- Table 3:
    - Following from the previous point, I feel Pass@1 is an unfair metric for comparison. If we focus on Cons@32, I don’t see a significant difference between +SFT and +LeaP.
    - The paper states:
    > We synthesize responses by applying LeaP to DeepSeek-R1-Distill-Qwen-32B (Guo et al., 2025) and
filter suitable responses using a rule-based selection mechanism. We use supervised finetuning to train
our LeaP-T models, starting from the 1.5B, 7B, and 14B versions of DeepSeek-R1-Distill-Qwen (Guo
et al., 2025).

      Based on this description, LeaP-T is effectively distilled from a 32B model, which is substantially larger than the 1.5B, 7B, and 14B models reported in Table 3. This makes the comparison not fair.


- The routing design appears rather ad hoc to me. In particular, the use of Levenshtein distance, a string-editing metric, seems unjustified, as it measures character-level differences without capturing semantic content, making it unclear how it meaningfully reflects the similarity or divergence of reasoning paths.

**Questions:**

see above.

---

### Official Review · Reviewer_pUaZ · 2025-10-31

**Soundness:** 1
**Presentation:** 2
**Contribution:** 1
**Rating:** 2
**Confidence:** 4

**Summary:**

The paper proposes LeaP, an aggregation method for reasoning over multiple reasoning traces. LeaP, different from simple majority voting, introduces a summarization module that summarizes the first T tokens of each trace (to reduce context) and then each summarized trace conditions on other traces to further continue the generation. Final answer is obtained by majority voting over these traces. Experiments are conducted with R1-distilled and QwQ models on different reasoning benchmarks to show that LeaP outperforms majority voting.

**Strengths:**

* LeaP introduces a simple method for learning from multiple reasoning traces. Experiments show that this can outperform majority voting.

* The authors conduct some analysis on how often the summarization module should be invoked and how many summaries should be aggregated.

**Weaknesses:**

* The biggest weakness of this paper is its lack of novelty. Reasoning over multiple samples (or meta-reasoning) is an old concept (e.g., see  [1], [2]) and here the only difference seems to be the summarizer module. I also don't see a clear ablation of this summarizer module to understand its usefulness. I also think that the method is not necessarily a multi-agent method because the experiments are limited to trajectories from the same underlying model. Related work also should not be in the appendix which makes it harder to understand the key contributions of the work.

* Some key details of the method as well as of the experimental setup are also missing from the paper. Since benchmarks like AIME are quite small, the paper should report the number of seeds and variance. Second, the aggregation prompt also seems to be missing. Third, assuming each LeaP block corresponds to one round of aggregation, how many rounds of aggregation are done? If it's > 1, I'd like to see a plot of baseline vs LeaP across rounds.

* The experiments are with relatively older models. To better understand the usefulness of the method on stronger models, it should be applied to newer Qwen3 models in both thinking/non-thinking modes at varying sizes. This will shed some light on how this method scales. Concurrent works have moved away from prompting-based aggregation to adopt RL-trained aggregation, casting doubt on the effectiveness of prompting-based aggregation [3].

[1] Learning to Reason Across Parallel Samples for LLM Reasoning. Qi et al. 2025

[2] Answering Questions by Meta-Reasoning over Multiple Chains of Thought. Yoran et al., 2023

[3] The Majority is not always right: RL training for solution aggregation. Zhao et al. 2025

**Questions:**

* Unless I have missed it, could you perform a clean ablation of the summarization module? For example, compare LeaP with only multiple rounds of aggregation.

* How many rounds/LeaP modules do you actually use? Where can I find a scaling plot?

* How many seeds did you use to conduct the experiments?

* How well does your method work with the stronger Qwen3 models?

---

### Official Review · Reviewer_7yVd · 2025-11-02

**Soundness:** 2
**Presentation:** 2
**Contribution:** 2
**Rating:** 4
**Confidence:** 2

**Summary:**

LeaP introduces a mechanism through which multiple agents (copies of the same model, but with differing prompts) decode thoughts in parallel and exchange messages regularly between them. The authors also trained a specialized model that achieves strong results compared to DeepSeek.

**Strengths:**

Some of the key strenghts:
1. The paper points out a widespread phenomena, that the first tokens (prefixes) can strongly influence the results up completely breaking a decoded sentence.

2. The paper introduces a test-time only modifications (with multiple agents), studying the setup under multiple models and benchmarks (AIME 2024/2025, AIMO 2025, GPQA, ZebraLogic)

**Weaknesses:**

Related to weaknesses, I would call out:
1. Limited novelty, very similar to ensembling methods. Due to prompting, it seems rather novel, however a single naive baseline with multiple reasoning traces and vote aggregation would probably clarify better the contribution.

2. Compute comparison would be necessary, given that multiple reasoning chains easily consume a lot of inference cost. The fine-tuning comparison does not strongly make the case either, reducing a 14B to 7B model is not particularly impressive.

3. It would be great to understand how the method approace with models fine-tuning.

**Questions:**

Please see weaknesses.

---

### Note · Authors · 2025-12-16

I have read and agree with the venue's withdrawal policy on behalf of myself and my co-authors.